# Not All Tokens Matter All The Time: Dynamic Token Aggregation Towards Efficient Detection Transformers

**Jiacheng Cheng** [1]  **Xiwen Yao** [1]  **Xiang Yuan** [1]  **Junwei Han** [1]

## Abstract

The substantial computational demands of detection transformers (DETRs) hinder their deployment in resource-constrained scenarios, with the encoder consistently emerging as a critical bottleneck. A promising solution lies in reducing token redundancy within the encoder. However, existing methods perform static sparsification while ignoring the varying importance of tokens across different levels and encoder blocks for object detection, leading to suboptimal sparsification and performance degradation. In this paper, we propose **Dynamic DETR** (**Dynamic** token aggregation for **DE**tection **TR**ansformers), a novel strategy that leverages inherent importance distribution to control token density and performs multi-level token sparsification. Within each stage, we apply a proximal aggregation paradigm for low-level tokens to maintain spatial integrity, and a holistic strategy for high-level tokens to capture broader contextual information. Furthermore, we propose center-distance regularization to align the distribution of tokens throughout the sparsification process, thereby facilitating the representation consistency and effectively preserving critical object-specific patterns. Extensive experiments on canonical DETR models demonstrate that Dynamic DETR is broadly applicable across various models and consistently outperforms existing token sparsification methods.

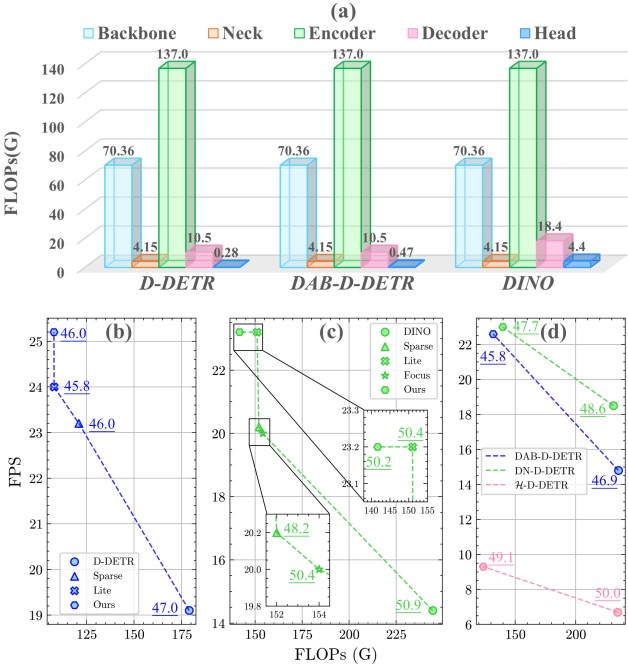

Figure 1: (a) Calculation distribution of D-DETR (Zhu et al., 2021), DAB-D-DETR (Liu et al., 2022), and DINO (Zhang et al., 2022), where the encoder emerges as the primary contributor to the overall computational load. The comparisons between various sparsification strategies on D-DETR (b) and DINO (c) in terms of FLOPs and FPS, while (d) exhibits the promotion of our method applying to DAB-D-DETR, DN-D-DETR (Li et al., 2022), and $\mathcal{H}$-D-DETR (Jia et al., 2023), where the AP of each method is underlined. All results are obtained using ResNet-50 as the backbone.

## 1. Introduction

**De**tection **Tr**ansformer (DETR) (Vaswani, 2017; Carion et al., 2020; Meng et al., 2021; Yao et al., 2021) has marked a transformative shift in object detection by incorporating

transformers. Deformable DETR (Zhu et al., 2021) further advances this paradigm with a multi-level deformable attention mechanism that accelerates convergence and boosts detection performance. Building on this foundation, various DETR-based models (Zhu et al., 2021; Liu et al., 2022; Li et al., 2022; Zhang et al., 2022) have been proposed, driving both the performance and versatility of object detection frameworks. However, the growing computational and memory demands of DETRs remain a significant obstacle to their practical deployment, particularly in resource-constrained environments that necessitate both efficiency

[1]School of Automation, Northwestern Polytechnical University, Xi'an, China. Correspondence to: Xiwen Yao <yaoxiwen@nwpu.edu.cn>.

*Proceedings of the $42^{nd}$ International Conference on Machine Learning*, Vancouver, Canada. PMLR 267, 2025. Copyright 2025 by the author(s).

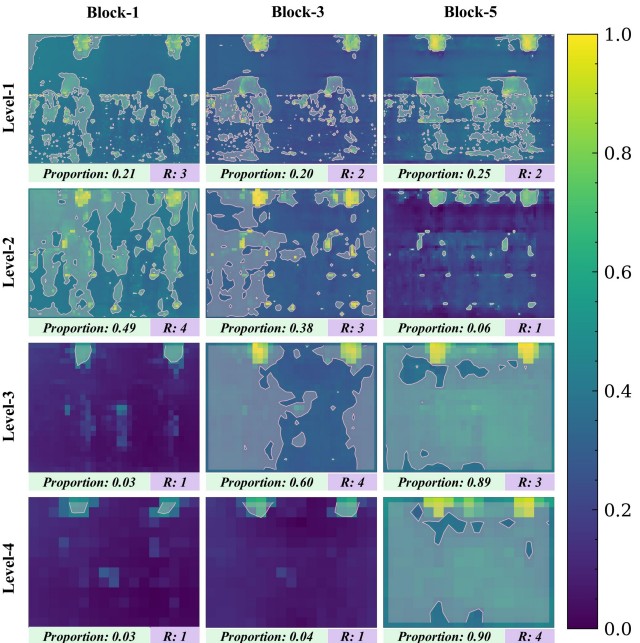

Figure 2: Visualization of attention weights of tokens across different encoder blocks. Noting that larger weights contribute more to the optimization and are deemed to be **important** in the attention mechanism according to pioneering efforts (Raganato & Tiedemann, 2018). We highlight the top 75% weights in the current block with pink-shaded mask, while the proportion of each level comes to ***Proportion***. Moreover, the term importance ranking ($R$) is introduced to intuitively represent the distribution of important tokens across different hierarchies. It can be seen that important tokens tend to **migrate** from shallow to deeper pyramidal levels as the depth of the encoder blocks increases.

and low-latency inference (Chen et al., 2022b; Zhang et al., 2023; Chen et al., 2022a).

To investigate the underlying computational burden, we analyze the floating point operations (FLOPs) for three prevalent DETR variants: Deformable DETR (Zhu et al., 2021), DAB DETR (Liu et al., 2022), and DINO (Zhang et al., 2022). As illustrated in Figure 1(a), the backbone and encoder contribute the most to the overall computational load. Although recent work has predominantly focused on optimizing the backbone (Rao et al., 2021; Chang et al., 2023; Liang et al., 2021; Long et al., 2023; Bolya et al., 2023), the encoder remains a significant bottleneck. This oversight is problematic, as efforts to reduce backbone complexity, such as with DynamicViT (Rao et al., 2021; Liu et al., 2024), lead to only modest reductions in FLOPs (approximately 12%), while causing a substantial drop in accuracy (4.6%).

Token sparsification has emerged as an effective solution for reducing the number of tokens based on their importance. To employ the importance of tokens, we visualize the atten-

tion weights across different levels and encoder blocks in Figure 2, and find that as we move deeper into the encoder blocks, the importance ranking $R$ of lower-level tends to decrease, while that of higher-level increases. This indicates that *token importance across levels is* **dynamic** *but gradually* **migrates** *from lower to higher levels as the encoder depth increases.* However, current approaches (Zheng et al., 2023; Wang et al., 2021; Roh et al., 2022), typically employ static, coarse-grained token discarding strategies based on global patches (Pu et al., 2022). Worse still, they resort to compensatory modules to mitigate information loss and fail to capture the hierarchical characteristic of tokens. Inspired by this insight, we propose Dynamic DETR, a framework that adapts token sparsification based on their evolving importance across encoder stages.

We apply token merging for token sparsification, which has been widely validated across the fields of transformers (Bolya et al., 2023; Liang et al., 2021; Long et al., 2023; Lee et al., 2024). Yet, its application in DETRs often overlook the diverse relationships between tokens and their respective levels, leading to suboptimal matching and significant performance degradation. Low-level tokens primarily capture detailed local information, with key features often concentrated in specific spatial regions (Cao et al., 2021). To preserve these details, merging should prioritize spatially proximal tokens, ensuring the retention of critical localized information. At higher levels, tokens with border receptive filed, related tokens may be distributed across different spatial locations (Lin et al., 2022). Holistic aggregation facilitates the direct integration of the most important tokens, without being constrained by local neighboring regions. To this end, we propose a strategy that applies distinct aggregation techniques across different levels, ensuring optimal token aggregation and preserving both local and global information effectively.

Furthermore, we hypothesize that sparse and non-sparse features fed into the detection heads should maintain an independent and identically distributed (*i.i.d.*) alignment (Gao et al., 2024). However, the varying inputs at encoder block introduce deviations from this ideal distribution. To address this issue, we propose a representational center-distance regularization to encourage the convergence toward the desired *i.i.d.* distribution.

To further validate the effectiveness and generality of Dynamic DETR, we evaluate it on COCO2017 using five DETR variants (Zhu et al., 2021; Liu et al., 2022; Zhang et al., 2022; Li et al., 2022; Jia et al., 2023). Results show that Dynamic DETR significantly reduces computational costs with minimal performance drops, outperforming other token sparsification methods (Figure 1). To sum up, the contributions of this paper are three-fold:

• Proposing Dynamic DETR, a dynamic token aggregation

pipeline for detection transformers that efficiently sparsifies tokens across encoder stages, achieving computation reduction with least performance loss.

• Devising a novel token aggregation strategy to handle multi-level token sparsification: a proximal aggregation paradigm for low-level tokens to preserve spatial details and a holistic merging strategy for high-level tokens to leverage their global receptive field. To the best of our knowledge, this is the first application of token aggregation techniques in detection transformers.

• Introducing the representation center-distance regularization that aligns the centroids of token distributions before and after sparsification. This improves feature consistency and enhances detection performance.

## 2. Related Work

### 2.1. Advancing Detection Transformer Designs

Recent advancements in Detection Transformers (DETRs) have aimed at improving detection accuracy and reducing computational costs.

On the performance front, Deformable DETR (Zhu et al., 2021) introduced multi-scale deformable attention to enhance convergence and accuracy, while Anchor DETR (Wang et al., 2022) and DAB-DETR (Liu et al., 2022) incorporated anchor-based priors to bridge the gap with traditional detectors. DN-DETR (Li et al., 2022) and DINO (Zhang et al., 2022) leveraged noisy ground-truth boxes to improve bounding box prediction. Despite these improvements, their high computational cost remains a barrier for real-world deployment.

Efficiency-focused DETR variants address this challenge. Efficient DETR (Yao et al., 2021) reduced encoder layers by initializing decoder queries with priors, while Lite DETR (Li et al., 2023) simplified token levels. Sparse approaches like Sparse DETR (Roh et al., 2022), PnP DETR (Wang et al., 2021), and Focus DETR (Zheng et al., 2023) optimized token updates by scoring salient tokens. However, these methods often rely on auxiliary modules and loss terms, complicating integration and limiting generality.

### 2.2. Token Sparsification Algorithms

Token sparsification has become central to accelerating vision transformers (ViTs) (Dosovitskiy et al., 2020), primarily comprising pruning-based and merging-based methods.

Pruning-based methods predict token importance scores using auxiliary modules. For instance, DynamicViT (Rao et al., 2021) and STViT-R (Chang et al., 2023) rank tokens based on their relevance, while A-ViT (Yin et al., 2022) uses an accumulative mask to discard tokens below a threshold.

While effective, these methods impose substantial computational overhead, rendering them less suitable for deployment in detection transformers.

Merging-based methods simplify computations by consolidating similar tokens. ToMe (Bolya et al., 2023) and CTS (Lu et al., 2023) merge tokens based on similarity, while EViT (Liang et al., 2021) and Long *et al.* (Long et al., 2023) use attention scores to aggregate discarded tokens into "super tokens". MCTF (Lee et al., 2024) extends merging by integrating multi-criteria for better information preservation. However, existing merging methods often lack hierarchical adaptability, and overly simplistic strategies risk entangling critical information.

## 3. Method

We propose a general paradigm for token sparsification in Detection Transformer (DETR) models, compatible with a wide range of architectures, including Deformable DETR (Zhu et al., 2021), DAB-DETR (Liu et al., 2022), DINO (Zhang et al., 2022), and other prominent variants (Li et al., 2022; Jia et al., 2023). Based on the analysis presented in Section 1, our method applies token merging selectively at specific stages within the encoder, as illustrated in Figure 3. Detailed descriptions of the proposed methodology are provided in the following subsections.

### 3.1. Preliminaries

Deformable DETR (Zhu et al., 2021) significantly enhances the performance of DETR (Carion et al., 2020) and has served as the foundation for several subsequent variants (Liu et al., 2022; Li et al., 2022; Zhang et al., 2022; Jia et al., 2023). For clarity and reproducibility, we adopt Deformable DETR as the baseline to describe our method.

Let $\{\boldsymbol{x}_l\}_{l=1}^{L}$ represent the multi-level input feature maps, where $\boldsymbol{x}_l \in \mathbb{R}^{C \times H_l \times s_l}$ denotes the representation at level $l$, with $H_l$, $s_l$, and $C$ corresponding to the height, width, and hidden states dimension, respectively. To meet the input requirements of the DETR encoder, each feature map is flattened into a sequence of embeddings $\boldsymbol{Z}_l \in \mathbb{R}^{N_l \times C}$, where $N_l = H_l \times s_l$ denotes the total number of tokens at level $l$, which are subsequently fed into the encoder.

The total number of tokens across all levels is $N = \sum_{l=1}^{L} N_l$. The computational complexity of encoder blocks is primarily dominated by the self-attention mechanism. In the case of the deformable multi-scale attention module, the complexity is $\mathcal{O}(2N_q C^2 + \min(NC^2, N_q K C^2))$, where $N_q$ is the number of tokens updated in the current block and $K$ is the number of sampled keys (Yang et al., 2021). Thus, the primary issue for decreasing the overall computational cost lies in the reduction of $N_q$.

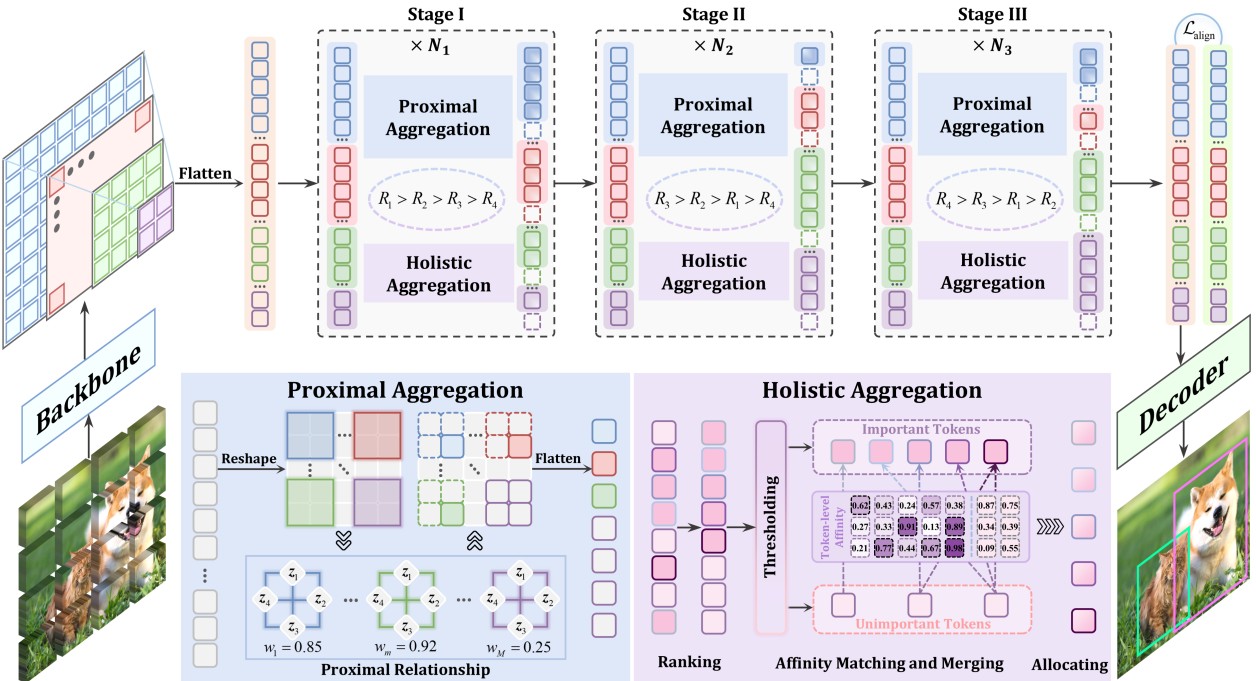

Figure 3: **Dynamic Token Aggregation Pipeline for Detection Transformers (*Dynamic DETR*).** The 6 encoder blocks are divided into three stages with $N_1 = 2$, $N_2 = 3$, and $N_3 = 1$. At each stage, tokens are merged based on dynamic importance ranking $\mathcal{R}^s$ of different levels. At lower levels, a **proximal aggregation** paradigm is employed, which first maps tokens to there original spatial grids and aggregates them within local windows. (*e.g.*, at level-2 with a window size of $n = 2$, each window contains 4 tokens.) Then the proximal relationship, denoted as $w_m$, is quantified to inform the aggregation decision. At higher levels, we devise a **holistic aggregation** strategy, where tokens are divided into important ones and unimportant ones according to their importance scores. Then unimportant tokens are aggregated into its most affine counterparts with the proposed affinity matching and aggregation mechanism. Finally, the sparsified features are constrained by $\mathcal{L}_{\text{align}}$ to preserve their fidelity by aligning the representation centroids between the non-sparsified and sparsified feature spaces.

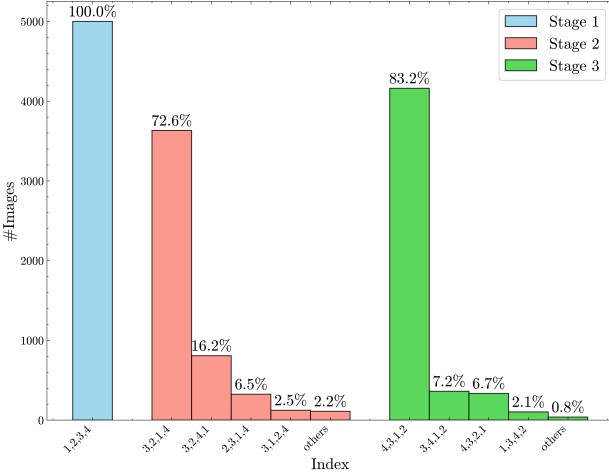

Figure 4: The distribution of importance ranking index (namely $\mathcal{I}$) across various pyramidal levels at three stages. Note that the statistics are captured on the COCO `val-set` images (Lin et al., 2014).

## 3.2. Dynamic Token Aggregation

The core of our aggregation mechanism lies in the dynamic evolution of the importance criterion for tokens across various hierarchical features and encoder blocks. Therefore, before delving into the proposed paradigm, we first investigate how the importance shifts within the encoder architecture.

**Notation.** Hereafter, we use the following notation for clarity and conformity. Given the importance ranking at $s$-th stage $R_{i_1} > R_{i_2} > \cdots > R_{i_L}$, we first introduce ranking index as follows:

$$\mathcal{I}^s = (i_1, i_2, \ldots, i_L), \text{where } 1 \leq i_j \leq L, i_j \neq i_k \ (j \neq k). \tag{1}$$

Then, the aforementioned ranking can be formulated with:

$$\mathcal{R}^s = (R_{i_1}, R_{i_2}, \ldots, R_{i_L}) \in \mathbb{R}^L. \tag{2}$$

Firstly, we divide the conventional six encoder blocks in the pipeline into three stages with $N_1 = 2$, $N_2 = 3$, and $N_3 = 1$. Then, we procure the distribution of importance ranking for each level $l \in \{1, \ldots, L\}$ at three stages in Figure 4. It is evident that the importance rankings are distributed

differently across the various encoder stages. In particular, during the first stage, we observe $\mathcal{R}^1 = (R_1, R_2, R_3, R_4)$, indicating that the model initially places greater emphasis on lower-level feature maps. As the stage depth increases, however, higher-level representations—such as those from levels $l = 3$ or $l = 4$—start to account for more of the important tokens, demonstrating a clear **migration** of importance. This finding aligns precisely with our observations in the Introduction (see Figure 2): namely, that the criterion for token-level importance within the encoders is **dynamic** and **evolving** across different stages.

Inspired by this observation, we therefore propose an dynamic token aggregation mechanism to dynamically reduce the number of tokens updated at each stage, thereby substantially diminishing the computational complexity while maintaining competitive performance. In contrast to pioneering works (Wang et al., 2021; Roh et al., 2022; Zheng et al., 2023) that fix a uniform retention ratio across all levels, we leverage statistic-based importance ranking to dynamically derive this key parameter for token sparsification at each level, thereby more accurately accommodating the evolving distribution of critical tokens within the encoder. To be specific, we first initialize the retention ratios for $L$ pyramidal levels with $\boldsymbol{\rho} = (\rho_1, \rho_2, \ldots, \rho_L) \in \mathbb{R}^L$. Then for stage $s$, its retention ratio $\boldsymbol{\rho}_s$ is formulated by reordering $\boldsymbol{\rho}$ based on the ranking index $\mathcal{I}^s$, namely:

$$\boldsymbol{\rho}^s = \boldsymbol{\rho}[\mathcal{I}^s] = (\rho_{i_1}, \rho_{i_2}, \ldots, \rho_{i_L}). \tag{3}$$

In this way, the number of tokens after sparsification for level $l$ is derived as $K_l = N_l \times \boldsymbol{\rho}_l^s$. To go a step further, the static ratio setting can be viewed as a special case of our dynamic strategy, wherein $\rho_1 = \rho_2 = \cdots = \rho_L$. To sum up, by reducing the number of tokens at each stage in an evolving manner, our approach achieves substantial computational savings, allowing more resources to be dedicated to processing the most critical token interactions

### 3.3. Multi-level Token Aggregation

Our solution is grounded in the observation that the distribution of important tokens across different hierarchies of Feature Pyramid Network (FPN) (Lin et al., 2017) progressively shifts as the encoder blocks deepen. Therefore, the sparsification process is dynamically executed through a multi-level framework, incorporating two complementary aggregation paradigms: (a) **Proximal Aggregation** ($g_p$) applied to low-level tokens ($l \leq L - 1$), which groups and merges tokens within spatially proximal patches. This paradigm not only preserves spatial coherence but also retains the fine-grained details that are essential for discriminative representation. (b) **Holistic Aggregation** ($g_h$) applied to ($l = L$) tokens, which leverages a global perspective for aggregation. It efficiently integrates long-range contextual patterns, thereby facilitating compact and semantically rich features. Gen-

erally, the aggregation of tokens at level $l$ is formalized as:

$$\hat{\boldsymbol{Z}}_l = \begin{cases} g_p(\boldsymbol{Z}_l) & \text{if } l \leq L - 1 \\ g_h(\boldsymbol{Z}_l) & \text{otherwise,} \end{cases} \tag{4}$$

where $\hat{\boldsymbol{Z}}_l$ illustrates the aggregated tokens at level $l$.

**Low-level: Proximal Aggregation.** The idea behind this paradigm is simple: tokens at lower pyramids are spatially abundant but semantically sparse, hence aggregating semantically similar proximal tokens builds a path to reduce the overall redundancy. Enlightened by previous works (Liu et al., 2021; Zhang et al., 2024), we introduce a window-based setup to achieve the optimal balance between compression efficiency and semantic integrity.

To be specific, we first map the tokens at level $l$, $\boldsymbol{Z}_l$, back to the original spatial dimensions $C \times H_l \times s_l$. Then, it will be partitioned into $M$ local patches with each containing $n \times n$ tokens, where $n = 2^{l-1}$ stands for the window size at level $l$. Intuitively, such setup fully leverages the resolution relationships between adjacent feature maps in the FPN. And most importantly, it is hyperparamter-free.

Next, to better investigate the local similarity of these patches, we capture the proximal relationship matrix for each patch, based on a fully connected bipartite graph constructed among tokens within the patch, which is formulated as the edge matrix $\mathbf{E}_m \in \mathbb{R}^{n \times n}$, where $m \in \{1, \ldots, M\}$. Concretely, supposing $\boldsymbol{z}_i$ and $\boldsymbol{z}_j$ belong to the identical proximal patch, then the edge weight $e_{i,j}$ between these two nodes is demonstrated as follows:

$$e_{i,j} = \frac{\boldsymbol{z}_i \cdot \boldsymbol{z}_j}{\|\boldsymbol{z}_i\|_2 \cdot \|\boldsymbol{z}_j\|_2}, \tag{5}$$

where $\boldsymbol{z}_i, \boldsymbol{z}_j \in \mathbb{R}^{C \times 1}$ (with $i, j \in \{1, \ldots, N_l\}$ and $i \neq j$) represent the $i$-th and $j$-th tokens of $\boldsymbol{Z}_l$, while $\cdot$ stands for the inner product, and $\|\cdot\|_2$ is the Euclidean norm.

To comprehensively quantify the semantic coherence of all proximal patches at level $l$, we capture the average weight sequence $\boldsymbol{W}_l = (w_1, w_2, \cdots, w_M)$, in which:

$$w_m = \frac{1}{n \times n} \sum_{i=1}^{n} \sum_{j=1}^{n} e_{i,j}. \tag{6}$$

Next, we rank $\boldsymbol{W}_l$ with decreasing order and filter top-$M'$ by indexing the set of selected indices as follows:

$$\mathcal{S}_l = \{\pi(1), \pi(2), \ldots, \pi(M')\}, \tag{7}$$

in which $\pi = \text{argsort}(\boldsymbol{W}_l, \text{descending})$. To go a step further, the number of remaining tokens after sparsification is $M' + (M - M') \cdot n^2$, which is consistent with that $K_l = N_l \times \boldsymbol{\rho}_l^s$ (see Sec. 3.2). In this way, $M'$ essentially comes as follows:

$$M' = \frac{N_l \cdot (1 - \boldsymbol{\rho}_l^s)}{n^2 - 1} \tag{8}$$

**Algorithm 1** Affinity Matching and Aggregation.

---

**Input:** Important token set $\mathcal{V} = \{z_i \mid i = 1, 2, \ldots, K_l\}$; Unimportant token set $\mathcal{U} = \{z_i \mid i = 1, 2, \ldots, N_l - K_l\}$; Import-to-unimportant affinity matrix $\mathbf{A}_{\text{i2u}} \in \mathbb{R}^{(N_l - K) \times N_l}$

**Output:** Updated important tokens $\hat{Z}_l = \{\hat{z}_i \mid i = 1, 2, \ldots, K_l\}$

Initialize $\hat{z}_i \leftarrow z_i, \{i = 1, 2, \ldots, K_l\}$.

**for** $i = 1$ **to** $N_l - K_l$ **do**
    $\texttt{ids}, \texttt{affs} \leftarrow \text{SortReverse}(\mathbf{A}_{\text{i2u}}, i)$
    Delete $\texttt{ids[0]}, \texttt{affs[0]}$
    **for** $j = 1$ **to** $T$ **do**
        **if** $\texttt{ids}[j] \in \{i = 1, 2, \ldots, K_l\}$ **then**
            $z_{\texttt{ids}[j]} \leftarrow \text{Merge}(z_{\texttt{ids}[j]} \in \mathcal{V}, z_j \in \mathcal{U})$
        **end if**
    **end for**
**end for**

---

Thereafter, within each qualified patch, the $n^2$ tokens are merged to form a super token which replaces the bottom-right one of current patch. Then, the aggregated token set $\mathcal{P} = \{z_i \mid i = \pi(1) \cdot n^2, \pi(2) \cdot n^2, \ldots, \pi(M') \cdot n^2\}$, while the remaining unmerged patches are flattened into $\mathcal{T}$. The final output tokens include both the aggregated super tokens and unmerged ones which can be formulated as follows:

$$\hat{Z}_l = \mathcal{P} \cup \mathcal{T}. \tag{9}$$

Thanks to the rigorous evaluation of semantic integrity across all patches, our proximal aggregation strategy ensures the integration of semantically meaningful tokens, thereby achieving an optimal balance between computational efficiency and representational quality.

**High-level: Holistic Aggregation.** Inspired by the bipartite soft matching mechanism (Bolya et al., 2023), we propose a tailored strategy to aggregate high-level tokens from a holistic fashion.

Given high-level tokens $Z_l = \{z_i \mid i = 1, 2, \ldots, N_l\}$, we first construct the affinity matrix $\mathbf{A} \in \mathbb{R}^{N_l \times N_l}$ by capturing the cosine similarity between all tokens with Eq. 5. Subsequently, the importance score for the $i$-th token is defined as follows:

$$\gamma_i = \frac{1}{N_l} \sum_{j=1, j \neq i}^{N_l} \frac{1}{e_{i,j}}. \tag{10}$$

To go a step further, tokens exhibiting lower similarity to others are assigned higher importance scores, as they are presumed to encapsulate more distinctive information. Next, we rank the importance sequence in descending order. According to Eq. 3, the top $K_l$ tokens are designated as important tokens $\mathcal{V} = \{z_i \mid i = 1, 2, \ldots, K_l\}$, while the remaining $(N_l - K_l)$ tokens are identified as unimportant ones $\mathcal{U} = \{z_j \mid j = 1, 2, \ldots, N_l - K_l\}$.

To preserve token relationships for further processing, we employ an affinity matching strategy. Each unimportant token is paired with its most similar important tokens, and the corresponding visual patterns are aggregated into these counterparts. Then, the import-to-unimportant affinity matrix $\mathbf{A}_{\text{i2u}} \in \mathbb{R}^{(N_l - K_l) \times N_l}$ is derived from $\mathbf{A}$. For each unimportant token, the top-$T$ similar tokens are identified by selecting the second-largest similarity value from each row of $\mathbf{A}_{\text{i2u}}$. We conduct ablations to procure the optimal setup of $T$, and further details are provided in Sec. A.

After identifying the top-$T$ tokens for each unimportant token, we verify whether these tokens belong to the set of important tokens $\mathcal{V}$. If they do, the representation of the unimportant token is aggregated into its corresponding important tokens. Note such one-to-many matching strategy can optimally embed the information from tokens in $\mathcal{U}$. Moreover, the overall matching and aggregation pipeline is summarized in Algorithm 1.

This token merging process not only preserves the semantic richness of the discarded tokens but also enhances the expressiveness of the important tokens by enriching them with complementary information.

### 3.4. Representational Center-distance Regularization

Let $Z$ represent the standard forward-pass representation and $\hat{Z}$ the corresponding sparsified pattern. In an ideal scenario, the distribution of predictions based on representations between the pre- ($p_{\text{pre}}$) and post-sparsification ($p_{\text{post}}$) stages should exhibit minimal deviation, thereby effectively reducing sparsification-induced information loss. Unfortunately, this is infeasible in practical implementations due to inherent limitations.

To mitigate this thorny issue without introducing auxiliary burdens, we propose representational center-distance regularization, which aligns the centroids of sparsified and non-sparsified representations:

$$\mathcal{L}_{\text{align}} = \|\nu_{\text{pre}} - \nu_{\text{post}}\|_2, \tag{11}$$

in which $\nu_{\text{pre}} = \mathbb{E}_{p_{\text{pre}}}[f(Z)]$ and $\nu_{\text{post}} = \mathbb{E}_{p_{\text{post}}}[f(\hat{Z})]$ stand for the representation centroids of the non-sparsified and sparsified models, respectively. Then, the overall objectives can be formulated as follows:

$$\mathcal{L}_{\text{total}} = \mathcal{L}_{\text{det}} + \lambda \cdot \mathcal{L}_{\text{align}}, \tag{12}$$

where $\mathcal{L}_{\text{det}}$ is the detection loss of the original model and $\lambda$ balances the contributions of different loss terms which is empirically default to 0.1 in our experiments. This regularization promotes alignment between centroids, ensuring consistent representation quality across sparsification levels and enhancing both robustness and generalization.

Table 1: Comparison of Dynamic DETR with DETR variants and other efficient DETR algorithms on the COCO2017 `val-set`. DETR, DAB DETR, DN DETR, Conditional DETR and Anchor DETR use a single DC5 feature, while the remaining detectors in the comparison employ multi-level features. "D-DETR" refers to Deformable DETR. Noting the top-performing results are highlighted in **Red** while the second-best ones are in **Blue**.

| Model | Paradigm | Epochs | AP | $AP_{50}$ | $AP_{75}$ | $AP_S$ | $AP_M$ | $AP_L$ | Params(M) | FLOPs(G) | FPS |
|---|---|---|---|---|---|---|---|---|---|---|---|
| DETR (Carion et al., 2020) | | 500 | 43.3 | 63.1 | 45.9 | 22.5 | 47.3 | 61.1 | 41.0 | 187.0 | 11.2 |
| DAB DETR (Liu et al., 2022) | | 50 | 44.5 | 65.1 | 47.7 | 25.3 | 48.2 | 62.3 | 44.0 | 203.0 | - |
| DN DETR (Li et al., 2022) | | 50 | 46.3 | 66.4 | 49.7 | 26.7 | 50.0 | 64.3 | 44.0 | 202.0 | - |
| Conditional DETR (Yao et al., 2021) | | 108 | 45.1 | 65.4 | 48.5 | 25.3 | 49.0 | 62.2 | 44.0 | 195.0 | 11.5 |
| Anchor DETR (Wang et al., 2022) | | 50 | 44.2 | 64.7 | 47.5 | 24.7 | 48.2 | 60.6 | 37.0 | 171.5 | 25.0 |
| D-DETR (Zhu et al., 2021) | Original | 50 | 47.0 | 66.1 | 50.9 | 30.0 | 49.8 | 61.9 | 41.0 | 179.0 | 19.1 |
| | Sparse (Roh et al., 2022) | | 46.0 | 65.9 | 49.7 | 29.1 | 49.1 | 60.6 | 41.0 | $121.0_{\downarrow 32.4\%}$ | $23.2_{\uparrow 21.5\%}$ |
| | Lite (Li et al., 2023) | | 45.8 | 65.1 | 49.3 | 27.7 | 49.1 | 61.1 | 41.0 | $\mathbf{108.0}_{\downarrow 39.7\%}$ | $\mathbf{24.0}_{\uparrow 25.7\%}$ |
| | Dynamic | | 46.0 | 65.1 | 50.1 | 29.0 | 48.9 | 60.6 | 41.0 | $\mathbf{107.9}_{\downarrow 39.7\%}$ | $\mathbf{25.2}_{\uparrow 31.9\%}$ |
| DINO (Zhang et al., 2022) | Original | 36 | 50.9 | 68.9 | 55.3 | 34.6 | 54.1 | 64.6 | 47.0 | 244.5 | 14.4 |
| | Sparse (Roh et al., 2022) | | 48.2 | 65.9 | 52.5 | 30.4 | 51.4 | 63.1 | 47.0 | $152.0_{\downarrow 37.8\%}$ | $20.2_{\uparrow 40.2\%}$ |
| | Lite (Li et al., 2023) | | 50.4 | - | 54.6 | 33.5 | 53.6 | 65.5 | 47.0 | $\mathbf{151.0}_{\downarrow 38.2\%}$ | $\mathbf{23.2}_{\uparrow 61.1\%}$ |
| | Focus (Zheng et al., 2023) | | 50.4 | 68.5 | 55.0 | 34.0 | 53.5 | 64.4 | 48.0 | $154.0_{\downarrow 37.0\%}$ | $20.0_{\uparrow 38.9\%}$ |
| | Dynamic | | 50.2 | 69.2 | 54.7 | 33.6 | 53.4 | 64.4 | 47.0 | $\mathbf{141.7}_{\downarrow 42.0\%}$ | $\mathbf{23.2}_{\uparrow 61.1\%}$ |
| DAB-D-DETR (Liu et al., 2022) | Original | 50 | 46.9 | 66.0 | 50.8 | 30.1 | 50.4 | 62.5 | 44.0 | 235.4 | 14.8 |
| | Dynamic | | 45.8 | 64.4 | 49.7 | 29.0 | 49.3 | 60.2 | 44.0 | $\mathbf{131.8}_{\downarrow 44.0\%}$ | $\mathbf{22.6}_{\uparrow 52.7\%}$ |
| DN-D-DETR (Li et al., 2022) | Original | 50 | 48.6 | 67.4 | 52.7 | 31.0 | 52.0 | 63.7 | 48.0 | 231.3 | 18.5 |
| | Dynamic | | 47.7 | 66.5 | 51.8 | 30.3 | 50.6 | 62.6 | 48.0 | $\mathbf{139.8}_{\downarrow 39.6\%}$ | $\mathbf{23.0}_{\uparrow 24.3\%}$ |
| $\mathcal{H}$-D-DETR (Jia et al., 2023) | Original | 36 | 50.0 | - | - | 32.9 | 52.7 | 65.3 | 48.0 | 234.8 | 6.7 |
| | Dynamic | | 49.1 | 65.7 | 54.0 | 32.7 | 52.9 | 64.3 | 48.0 | $\mathbf{123.6}_{\downarrow 47.4\%}$ | $\mathbf{9.3}_{\uparrow 38.8\%}$ |

# 4. Experiments

## 4.1. Implementation Details

All experiments are implemented using the detrex framework (Ren et al., 2023). Training is performed on 4 Nvidia GPUs with a batch size of 2. To ensure fair comparisons, we strictly follow the configurations of baseline models, including hyperparameters, network architectures, and loss functions.

The primary experiments are performed on the COCO 2017 dataset, which contains 118K training images and 5K validation images. We adopt ResNet-50 as the backbone, and use 6 encoder blocks and 4 pyramid levels by default. To evaluate the generalizability of our approach, we additionally conduct experiments on LVIS v1.0 (Gupta et al., 2019) and PASCAL VOC 2007 (Everingham et al.) within the DINO framework. LVIS v1.0 (Gupta et al., 2019) presents a large-vocabulary setting with a long-tailed category distribution, challenging models to detect rare objects. In contrast, PASCAL VOC 2007 (Everingham et al.) is a smaller-scale dataset that focuses on simpler object layouts and limited training samples. Moreover, we evaluate our method with the Swin-T backbone (Liu et al., 2021), a hierarchical vision transformer, to assess its effectiveness.

## 4.2. Experimental Results

### 4.2.1. MAIN RESULTS

Comprehensive results are summarized in Table 1, contrasting DETR variants, efficient DETR algorithms, and the enhanced versions with our Dynamic DETR. When applied to Deformable DETR, Lite DETR reduces FLOPs by 58.0G (32.4%) with a minor 1.0% drop in accuracy. In comparison, Dynamic DETR achieves a more substantial reduction of FLOPs (↓39.7%), while outperforming Lite DETR with comparable FLOPs and an additional 0.2% gain in precision. Sparse DETR achieves the same accuracy but incurs more FLOPs (13.1G) than our method. For DINO, Dynamic DETR reduces the overall computational cost by 42.0%, with only a 0.7% drop in AP. In contrast, Lite DETR and Focus DETR reduce FLOPs by 38.2% and 37.0%, respectively, with slightly smaller performance drops (0.5% AP). Dynamic DETR achieves the best trade-off by offering a higher reduction in computational cost while maintaining competitive accuracy.

We further evaluate Dynamic DETR on three advanced DETR variants: DAB-D-DETR, DN-D-DETR, and $\mathcal{H}$-D-DETR. Dynamic DETR can reduce the models' computational cost by varying amounts (ranging from 39.6% to 47.4%), while only incurring a minimal performance loss (about 1.0%). Dynamic DETR demonstrates superior computational efficiency and accuracy across a wide range of DETR architectures. This, makes it an good choice for real-time and resource-constrained applications.

## 4.2.2. ADDITIONAL RESULTS

To further assess the generalizability and robustness of our approach, we conduct supplementary experiments across multiple datasets and architectures. All models are trained for 12 epochs with consistent settings unless otherwise specified.

**Results on LVIS v1.0.** Table 2 presents the results on LVIS v1.0, a large-scale benchmark featuring over 1200 categories with a long-tailed distribution (Gupta et al., 2019). Dynamic DINO achieves a strong trade-off between performance and efficiency. While Focus DINO yields the highest overall AP and $AP_r$, Dynamic DINO delivers a competitive AP of 23.4 and achieves the best $AP_f$. Notably, our method reports the lowest FLOPs and the highest FPS among all variants, highlighting its efficiency advantage in long-tailed detection scenario.

Table 2: Detection performance on LVIS v1.0.

| Model | AP | $AP_{50}$ | $AP_{75}$ | $AP_r$ | $AP_c$ | $AP_f$ | FLOPs(G) | FPS |
|---|---|---|---|---|---|---|---|---|
| DINO | 26.1 | 34.5 | 27.5 | 8.3 | 24.1 | 36.1 | 247.1 | 19.8 |
| Sparse DINO | 22.9 | 32.0 | 24.2 | 8.4 | 21.3 | 30.9 | 151.7 | 21.2 |
| Lite DINO | 20.2 | 28.0 | 21.4 | 3.0 | 17.5 | 30.8 | 160.0 | 16.0 |
| Focus DINO | **23.7** | **32.9** | **25.2** | **10.2** | **21.7** | 31.9 | 168.2 | 20.4 |
| Dynamic DINO | 23.4 | 31.8 | 25.0 | 7.7 | 20.8 | **33.4** | **146.6** | **22.5** |

**Results on PASCAL VOC 2007.** As shown in Table 3, we evaluate our method on PASCAL VOC 2007, a compact dataset with 20 object classes and relatively simple scenes (Everingham et al.). Dynamic DINO attains the highest mAP (63.8%) among all lightweight variants, while consuming only 135.2 GFLOPs. Compared to the original DINO, our method maintains competitive accuracy with just 56% of the computation, and significantly outperforms Sparse DINO and Focus DINO in both accuracy and speed. Despite its lightweight design, Lite DINO significantly underperforms compared to its counterparts, potentially due to the necessity of prolonged training to achieve satisfactory convergence. To sum up, the promising results further underscore the generality of our design in efficient object detection.

Table 3: Detection performance on PASCAL VOC 2007.

| Model | mAP | FLOPs(G) | FPS |
|---|---|---|---|
| DINO | 65.7 | 241.6 | 15.5 |
| Sparse DINO | 62.5 | 141.4 | 19.6 |
| Lite DINO | 38.1 | 151.0 | **21.3** |
| Focus DINO | 51.4 | 152.1 | 20.2 |
| Dynamic DINO | **63.8** | **135.2** | 21.1 |

**Results on Swin-T Backbone.** We also assess our method with Swin-T, a hierarchical vision Transformer (Liu et al., 2021), to examine its compatibility with modern backbones.

Table 4 shows that Dynamic DINO consistently surpasses other variants across most metrics. It achieves the lowest FLOPs and the highest FPS, validating that the dynamic token aggregation strategy scales well with multi-stage, patch-based architectures. Compared to Sparse DINO, Dynamic DINO improves AP by 0.3 points while reducing computational cost by over 9%, indicating a better accuracy-efficiency trade-off.

Table 4: Performance comparison using the Swin-T backbone.

| Model | AP | $AP_{50}$ | $AP_{75}$ | $AP_S$ | $AP_M$ | $AP_L$ | FLOPs(G) | FPS |
|---|---|---|---|---|---|---|---|---|
| DINO | 51.5 | 70.2 | 56.5 | 34.6 | 54.5 | 67.0 | 252.3 | 14.0 |
| Sparse DINO | 49.6 | 68.4 | 54.1 | 32.1 | 52.6 | 65.3 | 137.0 | 18.0 |
| Lite DINO | 48.3 | 66.1 | 52.8 | 30.3 | 51.6 | 64.0 | 151.0 | 16.8 |
| Focus DINO | 49.9 | 68.2 | 54.3 | **32.9** | 52.8 | 65.1 | 156.9 | 15.3 |
| Dynamic DINO | 49.9 | **68.8** | 54.3 | 32.8 | **52.9** | **65.4** | **149.4** | **18.2** |

## 4.2.3. ABLATION STUDIES

We perform all the ablation studies on DINO to evaluate the effectiveness of the proposed components. The results are summarized in Table 5.

**Effect of the Dynamic Token Aggregation Mechanism.** To assess the impact of the Dynamic Token Aggregation (DTA) module, we replace it with a static variant using a fixed token retention ratio of 0.3 across all pyramid levels. Under this setting, the overall AP achieved is 48.2%, which is 1.4 points lower than the full dynamic version. This performance gap highlights the limitations of static aggregation, which fails to adaptively prioritize token relevance, and demonstrates the effectiveness of our DTA strategy in selectively retaining informative tokens.

**Effect of Multi-level Token Aggregation.** We further investigate the role of Multi-level Token Aggregation (MTA). Disabling token aggregation across stages by randomly selecting and retaining tokens with dynamic ratios results in a significant performance drop to 46.6% AP, despite the reduction in computational cost to 134.7G FLOPs. Additionally, we assess the role of affinity matching within the holistic token aggregation process. Removing affinity matching yields an AP of 49.8%, while enabling it improves performance to 50.2%, with a minor increase of 1.5G FLOPs. These results demonstrate the effectiveness of hierarchical aggregation and highlight the advantages of incorporating semantic similarity matching in holistic token aggregation, achieving a balanced accuracy-efficiency trade-off.

**Effect of Representational Center-distance Regularization.** Finally, we evaluate the impact of the Representational Center-distance Regularization (RCDR). Incorporating RCDR results in a significant performance improvement of 0.6% AP (from 49.6% to 50.2%) with no additional computational overhead. This result indicates that RCDR

Table 5: Ablation studies of the proposed components on COCO 2017. We evaluate the individual and combined contributions of the Dynamic Token Aggregation (DTA), Multi-level Token Aggregation (MTA) with and without Affinity Matching (AM), and Representational Center-distance Regularization (RCDR).

| DTA | MTA | RCDR | AP | AP$_{50}$ | AP$_{75}$ | AP$_S$ | AP$_M$ | AP$_L$ | FLOPs(G) |
|---|---|---|---|---|---|---|---|---|---|
| ✓ | ✗ | ✗ | 46.6 | 68.3 | 50.1 | 29.9 | 49.9 | 63.1 | 134.7 |
| ✗ | ✓ (*with* AM) | ✗ | 48.2 | 64.5 | 52.9 | 30.9 | 52.1 | 63.1 | 140.6 |
| ✓ | ✓ (*w/o* AM) | ✓ | 49.8 | **69.9** | 54.2 | 33.1 | 53.4 | **65.4** | **140.2** |
| ✓ | ✓ (*with* AM) | ✗ | 49.6 | 68.1 | 54.2 | 32.0 | 52.9 | 65.2 | 141.7 |
| ✓ | ✓ (*with* AM) | ✓ | **50.2** | 69.2 | **54.7** | **33.6** | **53.4** | 64.4 | 141.7 |

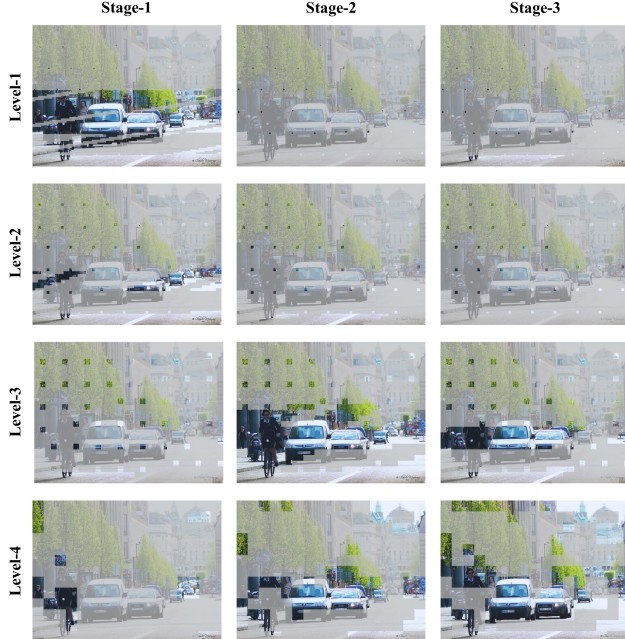

Figure 5: The distribution of important tokens, in which unimportant ones are masked.

#### 4.2.4. VISUALIZATION AND QUALITATIVE ANALYSIS

We visualize the distribution of important tokens across three stages in Figure 5. Consistent with our design, earlier stages retain a higher proportion of tokens at lower levels, while the focus gradually shifts toward higher levels in later stages. This demonstrates that critical information is effectively preserved throughout the merging process. At lower levels (*e.g.*, Levels 1–3), proximal tokens corresponding to spatially similar regions, are merged. In contrast, at higher levels, token aggregation is guided by the global semantic relationships. Notably, tokens with unique or distinctive features are preserved, reflecting the model's ability to prioritize salient information during the aggregation process.

## 5. Conclusion

In this work, we proposed Dynamic DETR (dynamic token aggregation for detection transformers) to tackle the computational bottlenecks inherent in DETRs. By leveraging token redundancy at multiple levels and across different encoder blocks, Dynamic DETR dynamically adjusts token representations while preserving essential semantic information. Through extensive experiments on various DETR variants, Dynamic DETR demonstrates state-of-the-art performance in terms of speed and accuracy, surpassing existing token sparsification competitors. Its flexibility allows it to be integrated seamlessly into diverse architectures, paving the way for efficient DETRs. Future work will explore adaptive sparsification strategies tailored to varying input characteristics and task requirements. Additionally, we aim to extend Dynamic DETR to other Transformer-based architectures, broadening its applicability to a wider range of tasks in computer vision and beyond.

## Acknowledgements

This work was supported in part by the National Natural Science Foundation of China under Grant 62471398 and Grant 62136007, in part by the National Science and Technology Major Project of China under Grant 2022ZD0119005.

## Impact Statement

This paper presents work whose goal is to advance the field of Machine Learning. There are many potential societal consequences of our work, none of which we feel must be specifically highlighted here.

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

# A. Discussions

To gain deeper insights into the effectiveness and robustness of our method, we conducted a series of experiments to analyze the impact of various hyperparameters and design choices.

**Choice of Token Retention Ratios.** We present a systematic investigation into the impact of dynamic token retention ratios on model performance through a comprehensive experimental framework. Our methodology evaluates retention ratio configurations by exploring an upper bound range of 0.4 to 0.8 and a lower bound range of 0.1 to 0.5, with detailed results documented in Table 6. As retention ratios increase, both computational complexity (measured in FLOPs) and model accuracy improve. For example, increasing the retention ratios from $(0.4, 0.3, 0.2, 0.1)$ to $(0.8, 0.7, 0.6, 0.5)$ results in a substantial 29.7% increase in computational requirements, with FLOPs rising from 107.9G to 140.0G for Deformable DETR. Concurrently, the AP metric exhibits a modest improvement (from 46.0% to 46.8%). This empirical evidence highlights a critical trade-off: while higher retention ratios consistently enhance the overall performance, the marginal gain in accuracy is disproportionately small compared to the significant increase in computational overhead. Based on a comprehensive analysis of these metrics, we propose an optimized configuration strategy that carefully balances token selection ratios for each model. This maximizes the efficiency-accuracy trade-off, ensuring optimal performance across all evaluation metrics while maintaining computational feasibility.

Table 6: Performance comparison of different token retention ratios $\rho$ on different models. We include the purple results in Table 1.

| $\rho$ | D-DETR (Zhu et al., 2021) | | DINO (Zhang et al., 2022) | | DAB D-DETR (Liu et al., 2022) | | DN D-DETR (Li et al., 2022) | | $\mathcal{H}$ D-DETR (Jia et al., 2023) | |
|---|---|---|---|---|---|---|---|---|---|---|
| | AP | FLOPs(G) | AP | FLOPs(G) | AP | FLOPs(G) | AP | FLOPs(G) | AP | FLOPs(G) |
| $(0.4, 0.3, 0.2, 0.1)$ | 46.0 | 107.9 | 49.9 | 132.6 | 45.4 | 120.1 | 47.5 | 128.4 | 49.1 | 123.6 |
| $(0.5, 0.4, 0.3, 0.2)$ | 46.3 | 114.7 | 50.2 | 141.7 | 45.8 | 131.8 | 47.7 | 139.8 | 49.2 | 135.3 |
| $(0.6, 0.5, 0.4, 0.3)$ | 46.5 | 122.9 | 50.5 | 159.3 | 46.1 | 146.2 | 47.9 | 154.5 | 49.4 | 149.6 |
| $(0.7, 0.6, 0.5, 0.4)$ | 46.6 | 131.5 | 50.6 | 174.5 | 46.2 | 161.4 | 48.2 | 169.6 | 49.5 | 164.8 |
| $(0.8, 0.7, 0.6, 0.5)$ | 46.8 | 140.0 | 50.8 | 189.5 | 46.3 | 176.4 | 48.3 | 184.7 | 49.7 | 179.8 |

**Choice of Options for Token Aggregation.** We demonstrate the optimal choices of token retention ratios on various models. For consistency and convenience in further discussions, all subsequent experiments are conducted on DINO (Zhang et al., 2022).

Table 7: Performance comparison across different experimental settings. All experiments are conducted on DINO. We include the purple results in Table 1 and Table 6.

| Method | AP | $AP_{50}$ | $AP_{75}$ | FLOPs(G) |
|---|---|---|---|---|
| Retain | 49.7 | 68.2 | 53.8 | 141.7 |
| Sum | 49.9 | 68.5 | 54.1 | 141.7 |
| Max | 50.0 | 69.6 | 54.5 | 141.7 |
| Mean | 50.2 | 69.2 | 54.7 | 141.7 |

(a): Different token aggregation strategies.

| $T$ | AP | $AP_{50}$ | $AP_{75}$ | FPS | FLOPs(G) |
|---|---|---|---|---|---|
| 1 | 50.1 | 69.5 | 54.5 | 25.2 | 141.7 |
| 3 | 50.2 | 69.2 | 54.7 | 23.3 | 141.7 |
| 5 | 50.2 | 69.4 | 54.6 | 18.4 | 141.7 |
| 7 | 50.2 | 69.6 | 54.6 | 12.1 | 141.7 |
| 9 | 50.3 | 69.6 | 54.6 | 7.9 | 141.7 |

(b): The number of aggregation tokens.

| $\lambda$ | AP | $AP_{50}$ | $AP_{75}$ | FLOPs(G) |
|---|---|---|---|---|
| 0.01 | 49.7 | 69.4 | 54.1 | 141.7 |
| 0.03 | 49.8 | 69.8 | 54.3 | 141.7 |
| 0.1 | 50.2 | 69.2 | 54.7 | 141.7 |
| 0.8 | 49.9 | 69.9 | 54.4 | 141.7 |
| 1 | 49.4 | 69.6 | 53.5 | 141.7 |

(c): Varying setups of the regularization coefficient $\lambda$.

As outlined in the Proximal Aggregation section, the tokens within each eligible patch are merged to form a super token, which is retained at the bottom-right corner of the current patch. Subsequently we examine how various aggregation

strategies affect model performance during unified representation generation, including: (a) Retain: Preserving the token located at the bottom-right corner of $n \times n$ region; (b) Sum: Summing all tokens within the current patch; (c) Max: Indexing the maximum value among all tokens within the current patch; (d) Mean: Calculating the average of all tokens within the current patch. The results summarized in Table 7(a) demonstrate that all three merging strategies outperform the non-merging baseline (Retain), highlighting the benefits of token aggregation for efficient modeling. Among these strategies, Mean achieves the best performance, with 50.2% AP, 69.2% $AP_{50}$, and 54.7% $AP_{75}$. We attribute the superior performance of Mean to its ability to evenly distribute information across tokens during aggregation. This approach preserves semantic consistency while reducing redundancy, making it particularly effective for compressing token representations without sacrificing discriminative power.

**The Number of the Most Similar Tokens in Holistic Aggregation.** We investigate the effect of varying the number of most similar tokens, $T$, set to 1, 3, 5, 7, and 9. The results, shown in Table 7(b), indicate that propagating information into a larger number of tokens generally enhances performance compared to using only one or three tokens. This improvement stems from the robustness of distributing information across multiple tokens. Specifically, aggregating into multiple tokens mitigates the impact of noise or erroneous information in individual tokens, as the other tokens are more likely to contain accurate and complementary information. However, the performance gains become marginal when $T$ exceeds 3. Additionally, increasing $T$ beyond this point causes a decline in inference speed, from 23.3 FPS to 7.9 FPS. Considering the trade-off between effectiveness and efficiency, we select $T = 3$ as the optimal setting for the number of the most similar tokens in our experiments.

**Coefficient of Representational Center-distance Regularization Term.** We analyze the effect of the regularization coefficient $\lambda$ in the representational center-distance regularization term on model performance. A range of $\lambda$ values, from 0.01 to 1, is tested, and the results are summarized in Table 7(c). The results show that $\lambda = 0.1$ achieves the best balance between representation alignment and model performance, resulting in an AP of 50.2%. When $\lambda$ is set too high (e.g., $\lambda = 0.8$ or 1), the model becomes over-regularized, focusing too heavily on aligning representation centroids at the cost of learning meaningful task-specific features, which leads to a decline in overall accuracy. On the other hand, when $\lambda$ is too low (e.g., $\lambda = 0.01$ or 0.03), the regularization effect is insufficient, leading to weak alignment between sparsified and non-sparsified representations, which results in suboptimal sparsification performance. Based on these observations, we select $\lambda = 0.1$ as the optimal value.

## B. Additional Visualization

Due to the application of our multi-level token aggregation strategy across various stages, direct visual comparison with existing coarse-grained token discarding approaches poses a challenge. To ensure a fair and interpretable evaluation, we visualize the top 300 decoder queries extracted from the encoder outputs of the vanilla DINO model (Zhang et al., 2022), Sparse DETR (Roh et al., 2022), and our proposed Dynamic DETR. These queries are directly responsible for the detection results, as illustrated in Figure 6.

In Figure 6, selected decoder queries are visualized as yellow regions, while unselected tokens are marked in purple. Sparse DETR predominantly focuses on inner areas of the most salient objects, effectively isolating them. However, it neglects background regions and other objects that are critical for accurate detection, particularly at higher levels. In contrast, our proposed Dynamic DETR incorporates a more comprehensive approach by preserving the intrinsic relationships among tokens, leading to a token distribution that more closely resembles the original DINO. This alignment ensures that both salient objects and essential contextual information are retained, enhancing the robustness and accuracy of the detection process.

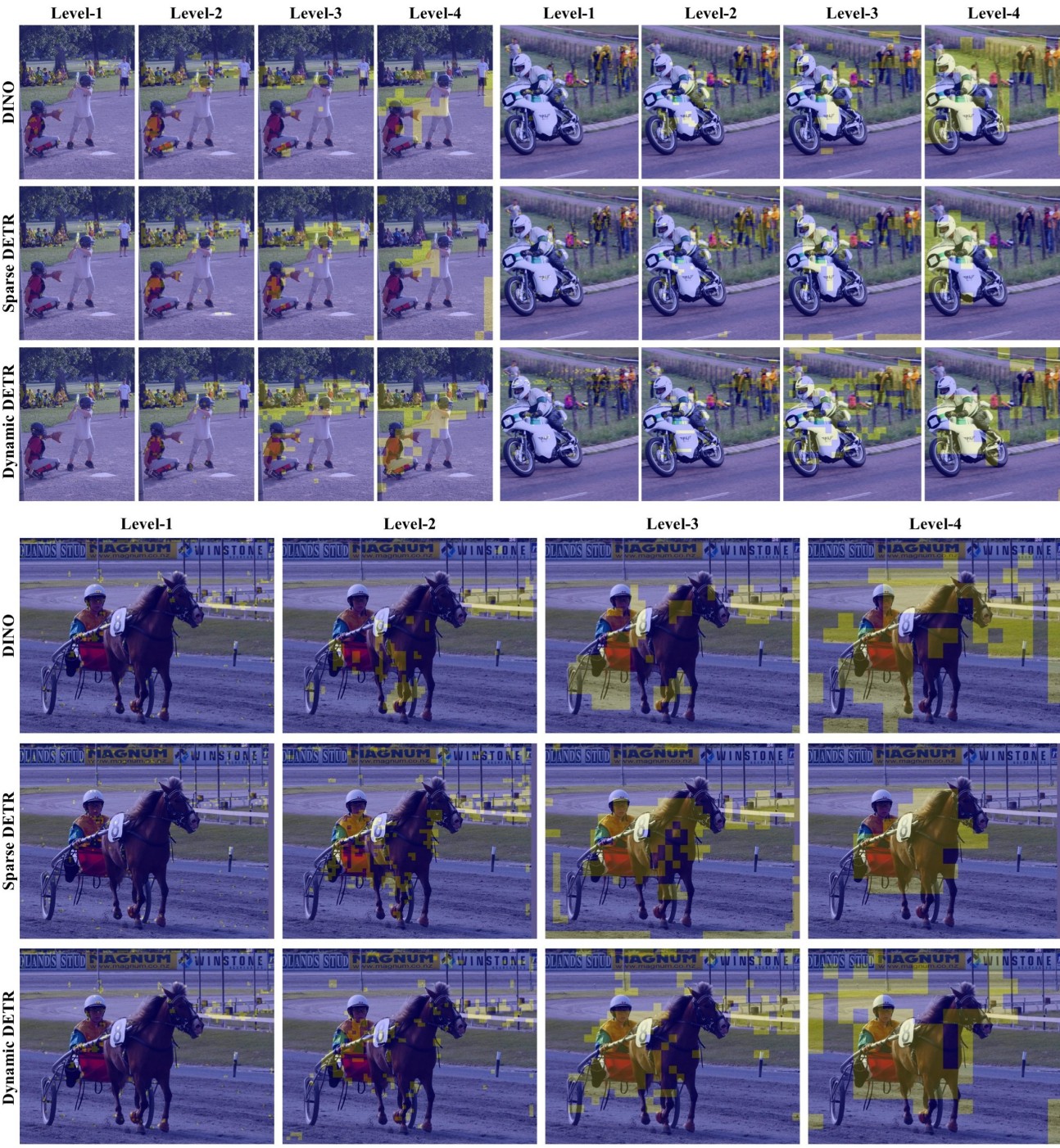

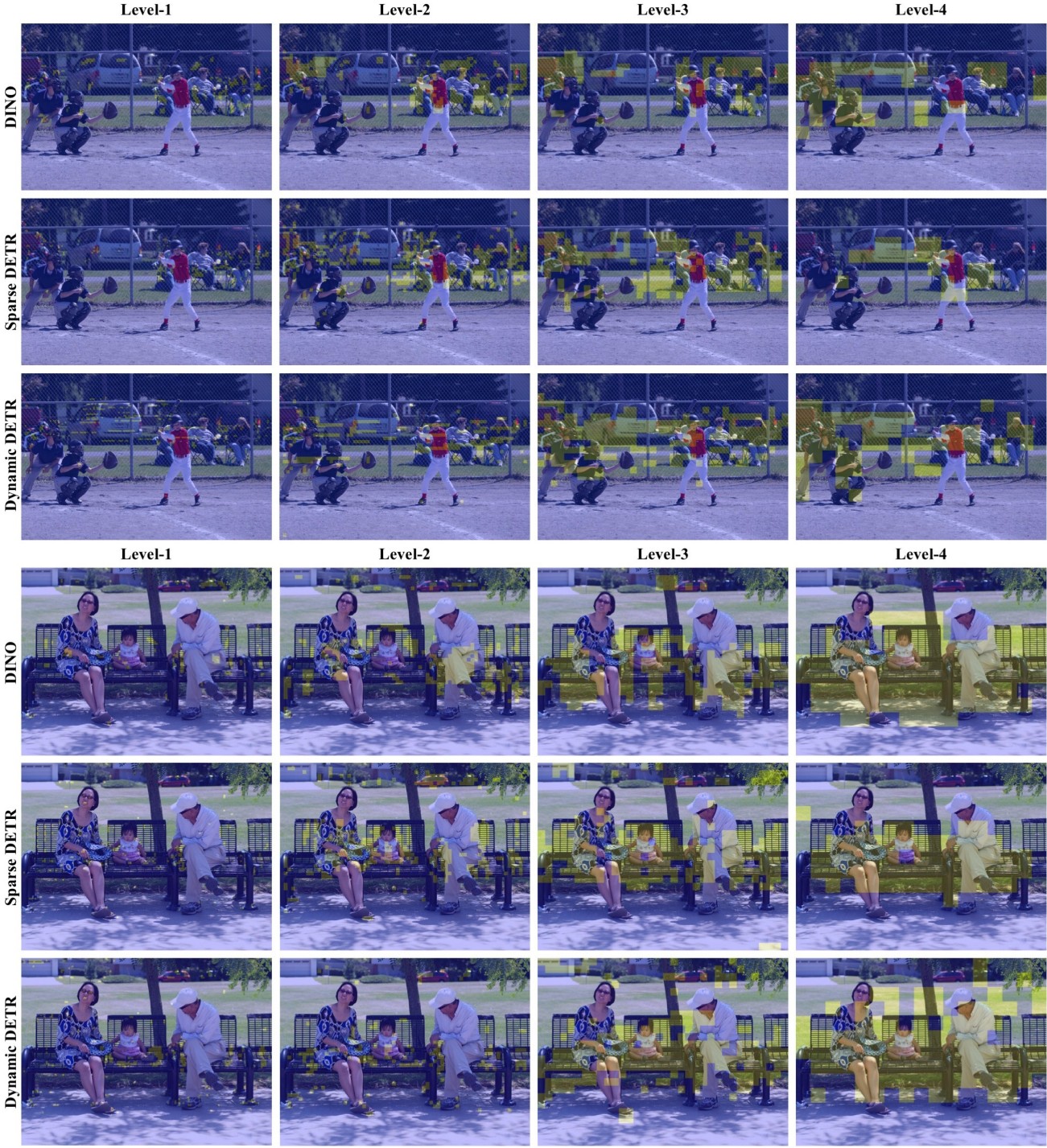

Figure 6: Visualization of the Top-300 decoder queries from the encoder output. Note the selected queries are highlighted with yellow.

