# OpenReview forum: "Not All Tokens Matter All The Time: Dynamic Token Aggregation Towards Efficient Detection Transformers"
_ICML.cc/2025/Conference — ICML 2025 poster_

### Official Review · Reviewer_naQi · 2025-03-11

**Overall Recommendation:** 3

**Summary:**

This paper proposes Dynamic DETR to reduce token redundancy within the encoder for improving the efficiency of DETR-like object detectors. This problem has also been studied by previous work such as Sparse DETR and Focus-DETR. Compared to these existing efforts, this work proposes a finer-grained sparsification approach, using different retention ratios for different feature pyramid levels and designing different sparsification strategies for different levels. Another improvement is the introduction of an additional loss function to distill the feature representation of the original encoder. The proposed method was validated on multiple DETR variants and demonstrated advantages over established methods.

**Claims And Evidence:**

The claim that “the distribution of important tokens across different hierarchies” follows a certain pattern has been validated in only one sample (Figure 2). Is this phenomenon universal? It is better to define some quantitative indicator, which would increase the credibility of this claim.

**Essential References Not Discussed:**

No, the existing related works have been well discussed

**Experimental Designs Or Analyses:**

The experimental designs are generally sound.

**Methods And Evaluation Criteria:**

Yes, the evaluation makes sense.

**Other Comments Or Suggestions:**

see the weaknesses part above

**Other Strengths And Weaknesses:**

Strengths:
1. The motivation makes sense. The proposed method uncovers the need to use different sparsification thresholds for different levels, which was neglected by previous works.
2. The experiments are adequate. The proposed method was validated on multiple DETR variants such as D-DETR, DINO and DAB-DETR.


Weaknesses:
1. There is some lack of clarity in the experimental setup.
- The paper does not indicate what device the FPS in Table 1 was measured on.
- For which backbone is Figure 1(a) obtained?
2. There are many writing issues:
- The figure captions are too long. It is recommended to move part of the content into the main text for better readability and clarity.
- Figure 1(a) provides limited information. It would be more informative to include a comparison of FLOPs before and after applying the proposed method.
- In Figure 1(b), both the horizontal (FLOPs) and vertical (FPS) axes represent efficiency. To enhance clarity and intuitiveness, it is recommended to replace one of the axes with AP.

**Questions For Authors:**

To what extent does the proposed method increase training costs? How does this compare to other counterparts?

**Relation To Broader Scientific Literature:**

No, not related to the broader scientific literature

**Theoretical Claims:**

No, it's not a theoretical paper.

---

> ### Author Rebuttal · Authors · 2025-04-01
>
> We appreciate your valuable suggestions, and next we respond to each comment as follows.
>
> ## W1: There is some lack of clarity in the experimental setup.
>  - *The paper does not indicate what device the FPS in Table 1 was measured on.*
>  - *For which backbone is Figure 1(a) obtained?*
>
> **Response**: Sorry for our unclear descriptions about the experimental setup.
>  - The FPS of all models in both the initial submission and the rebuttal phase is measured on a single RTX 3090, with the server load kept consistent.
>  - The backbone for Figure 1(a) is ResNet-50.
>
> ## W2: There are many writing issues.
>  - *The figure captions are too long. It is recommended to move part of the content into the main text for better readability and clarity.*
>  - *Figure 1(a) provides limited information. It would be more informative to include a comparison of FLOPs before and after applying the proposed method.*
>  - *In Figure 1(b), both the horizontal (FLOPs) and vertical (FPS) axes represent efficiency. To enhance clarity and intuitiveness, it is recommended to replace one of the axes with AP.*
>
> **Response**: We sincerely appreciate your detailed comments on writing and figure clarity.
>  - Regarding the figure captions, we will refine them by moving excessive details into the main text to improve readability.
>  - For Figure 1(a), we acknowledge its limited informativeness and will incorporate a FLOPs comparison before and after applying the proposed method to intuitively showcase the effectiveness of our approach.
>  - In Figure 1(b), we will revise the visualization to present Params vs. AP, providing a more balanced perspective on both efficiency and accuracy.
>
> Thanks again for your insightful suggestions, and we  will incorporate these modifications to enhance the overall clarity and readability.
>
> ## Q1: To what extent does the proposed method increase training costs? How does this compare to other counterparts?
>
> **Response**: Following your insightful suggestions as well as the recommendations from previous reviewers, we take DINO as the baseline detector and compare the training costs of our method and other competitors. Specific results are as follows.
>
> Tab. T1. Training cost comparisons between Ours (Dynamic DINO)  and other efficient solutions with ResNet-50 on the LVIS val-set, where the memory are captured when batch size=2.
> | Model | AP  | AP$_{\mathrm{50}}$  | AP$_{\mathrm{75}}$  | AP$_{\mathrm{r}}$  | AP$_{\mathrm{c}}$  | AP$_{\mathrm{f}}$   | FLOPs (G) | FPS  | GPU Memory (G) |
> |-|:-:|:-:|:-:|:-:|:-:|:-:|:-:|:-:|:-:|
> | DINO| 26.1 | 34.5 | 27.5 | 8.3 | 24.1 | 36.1 | 247.1 | 19.8 | 24244|
> | Sparse DINO| 22.9 | 32.0 | 24.2 | 8.4 | 21.3 | 30.9 | 151.7 | 21.2 | 22680|
> | Lite DINO| 20.2 | 28.0 | 21.4 | 3.0 | 17.5 | 30.8 | 160.0 | 16.0 | 42348|
> | Focus DINO| **23.7** | **32.9** | **25.2** | **10.2** | **21.7** | 31.9 | 168.2 | 20.4 | 22640|
> | Dynamic DINO| 23.4 | 31.8 | 25.0 | 7.7 | 20.8 | **33.4** | **146.6** | **22.5**| **22557**|
> ||
>
> Tab. T2. Training cost comparisons between Ours (Dynamic DINO) and other efficient solutions with  Swin-Transformer on the COCO val-set, where the memory are captured when batch size=2.
> | Model | AP  | AP$_{\mathrm{50}}$  | AP$_{\mathrm{75}}$|FLOPs (G) | FPS  | GPU Memory (G) |
> |-|:-:|:-:|:-:|:-:|:-:|:-:|
> |DINO|51.5|70.2|56.5|252.3|14.0|27892|
> |Sparse DINO|49.6|68.4|54.1|137.0|18.0|**24696**|
> |Lite DINO|48.3|66.1|52.8|151.0|16.8|26371|
> |Focus DINO|**49.9**|68.2|54.3|156.9|15.3|31933|
> |Dynamic DINO|**49.9**|68.8|54.3|**149.4**|**18.2**|27764|
> ||
>
> Tab. T3. Training cost comparisons between Ours (Dynamic DINO)  and other efficient solutions with  MobileNet-V2 on the COCO val-set, where the memory are captured when batch size=2.
> |Model|AP|AP$_{\mathrm{50}}$|AP$_{\mathrm{75}}$|FLOPs (G)|FPS|GPU Memory (G) |
> |-|:-:|:-:|:-:|:-:|:-:|:-:|
> |DINO|25.9|38.7|27.5|172.6|24.0|15304|
> |Sparse DINO|19.8|33.8|20.7|67.5|27.2|15810|
> |Lite DINO|18.8|29.2|19.9|78.4|**36.2**|33950|
> |Focus DINO|20.0|31.8|20.7|79.8|24.7|14141|
> |Dynamic DINO|**21.7**|**33.1**|**23.1**|**61.5**|36.0|**14082**|
> ||
>
> From the above results, it can be observed that the proposed Dynamic Token Sparsification strategy generally maintains a similar or even lower training cost compared to the baseline detector. In comparison to other counterparts, our method demonstrates clear advantages in both accuracy and speed.

---

### Official Review · Reviewer_2gMK · 2025-03-12

**Overall Recommendation:** 4

**Summary:**

This paper introduces Dynamic DETR designed to enhance the computational efficiency of DETR-based methods. The study identifies the encoder as the primary computational bottleneck and proposes a dynamic token sparsification strategy to reduce redundant tokens, effectively lowering computational complexity while preserving detection performance.

The proposed method features two key modules:

1. **Proximal Aggregation**: Merges tokens based on spatial adjacency to retain local structural details.
2. **Holistic Aggregation**: Ranks token importance and aggregates less important tokens into their most similar important counterparts, ensuring efficient representation.

Compared to the original DETR model, Dynamic DETR reduces computational cost by approximately 40% to 50% in terms of FLOPs, while only causing a minor 0.5% to 1% drop in AP.

**Claims And Evidence:**

The authors claim that the encoder is the main computational bottleneck in DETR models. The paper provides FLOPs analysis (Figure 1) showing that the encoder contributes significantly to the total computational cost.

In addition, the authors claim that token importance dynamically changes across different encoder stages and support this claim through statistical analysis of token importance distribution at various levels.

**Essential References Not Discussed:**

The paper provides a comprehensive discussion of previous related works.

**Experimental Designs Or Analyses:**

The paper mainly conducts experiments based on COCO2017. Although multiple DETR variants are tested, there is a lack of verification of generalization capabilities on other datasets (such as ADE20K, Cityscapes, etc.).

**Methods And Evaluation Criteria:**

Dynamic DETR incorporates Proximal Aggregation, Holistic Aggregation, and Center-distance Regularization to enhance token sparsification.

- Proximal Aggregation employs a window-based approach to determine proximity, ensuring that token merging preserves local structural integrity.
- Holistic Aggregation merges less important tokens into their most similar important tokens, effectively reducing redundancy while maintaining essential semantic information.
- Center-distance Regularization ensures that token representations remain statistically consistent before and after sparsification, preserving feature integrity and model stability.

**Other Comments Or Suggestions:**

1. Adding the corresponding symbols from the equations in the paper to Figure 3 would improve clarity and make it easier for readers to follow the method.

**Other Strengths And Weaknesses:**

**Strengths**

1. Compared to static sparsification methods, Dynamic DETR adopts a dynamic strategy, which more effectively balances computational cost and detection accuracy.
2. It is applicable to multiple DETR variants, including Deformable DETR, DINO, and DAB-DETR, demonstrating its broad compatibility.

**Weaknesses**

1. While Dynamic DETR reduces the overall FLOPs, the token importance computation and matching strategy introduce extra operations. A breakdown of this overhead in terms of FLOPs or latency would provide better clarity.
2. The paper primarily evaluates the method on COCO2017, lacking experiments on other datasets such as ADE20K and Cityscapes.
3. Compared to Lite DETR (Li et al., 2023), the improvements in AP are relatively small, despite similar FLOPs reduction. Given that Lite DETR achieves comparable performance, the advantages of Dynamic DETR in real-world applications should be further clarified.

**Questions For Authors:**

please refer to the Strengths And Weaknesses

**Relation To Broader Scientific Literature:**

This study focuses on designing a dynamic token sparsification strategy to address the encoder computation bottleneck in DETR. Unlike previous lightweight approaches, it introduces a more adaptive and efficient mechanism for token selection.

**Theoretical Claims:**

Not applicable, the article does not have proofs for theoretical claims.

---

> ### Author Rebuttal · Authors · 2025-04-01
>
> We really appreciate your constructive comments. We respond to each comment as follows.
>
> ## W1: A breakdown of token importance computation and matching strategy in terms of FLOPs or latency would provide better clarity
>
> **Response**: To quantify this overhead, we provide a detailed analysis of the FLOPs and Latency for these two components in **Tab. T1**.
>
> Tab. T1. FLOPs and latency analysis of our method based on DINO, and the results are based on COCO val-set.
> |Model|AP| AP$_{\mathrm{50}}$|AP$_{\mathrm{75}}$|FLOPs (G)|FPS|Latency (s)|
> |-|:-:|:-:|:-:|:-:|:-:|:-:|
> |DINO| 50.9 | 68.9 | 55.3|244.5|14.4|0.069|
> |w/o Token Imp. (Random Selecting)|46.6|68.3|50.1|134.7|27.8|0.036|
> |w/o Mat. (Prox Agg.&Holi Selecting)|49.8|69.9|54.2|140.2|24.8|0.040|
> |Dynamic DINO|50.2|69.2|54.7|141.7|23.2|0.043|
> ||
>
> When we discard the token importance in Eq. (10), namely randomly assigning tokens as important, the performance drops drastically. Meanwhile, turning the matching scheme off leads to a ~1% AP drop but improves FPS by 10.6. Moreover, both components contribute significantly to parameter reduction.
>
> ## W2: The paper primarily evaluates the method on COCO2017, lacking experiments on other datasets such as ADE20K and Cityscapes
>
> **Response**:
> To ensure fair comparisons, we follow the data setups of prior works (*Sparse detr, Lite detr, and Focus detr*), where COCO is primarily used for evaluation and analysis. ADE20K and Cityscapes are designed for segmentation tasks, making them less directly aligned with our study. Meanwhile, following the insightful advices from yours and previous reviewers, we conduct experiments on VOC and LVIS—two widely used object detection benchmarks beyond COCO.Note that all the models are with a ResNet-50 as the backbone and trained for 12 epochs.
>
> Tab. T2. Performance of DINO and various efficient solutions on the VOC2007 val-set.
> | Model| mAP|FLOPs (G)|FPS|
> |-|:-:|:-:|:-:|
> | DINO|65.7| 241.6|15.5|
> | Sparse DINO|62.5|141.4|19.6|
> | Lite DINO |38.1|151.0|21.3|
> | Focus DINO|51.4| 153.6|20.2|
> | Dynamic DINO|**63.8**|**135.2**|**21.1**|
> ||
>
> Tab. T3. Performance of DINO and various efficient solutions on the LVIS-1.0 val-set.
> |Model|AP|AP$_{\mathrm{50}}$|AP$_{\mathrm{75}}$|AP$_{\mathrm{r}}$|AP$_{\mathrm{c}}$|AP$_{\mathrm{f}}$|FLOPs (G)|FPS|
> |-|:-:|:-:|:-:|:-:|:-:|:-:|:-:|:-:|
> |DINO|26.1|34.5|27.5|8.3|24.1|36.1|247.1|19.8|
> |Sparse DINO| 22.9|32.0|24.2|8.4|21.3|30.9|151.7|21.2|
> |Lite DINO| 20.2 | 28.0|21.4|3.0|17.5|30.8|160.0|16.0|
> |Focus DINO| **23.7**| **32.9** | **25.2**|**10.2**|**21.7**|31.9|168.2|20.4|
> |Dynamic DINO| 23.4 | 31.8|25.0 |7.7|20.8|**33.4**|**146.6**|**22.5**|
> ||
>
> As exhibited in **Tab. T2** and **Tab. T3**, the results on VOC and LVIS datasets further showcase the superiority and generality of our dynamic strategy. Moreover, we plan to explore the potential of dynamic token sparsification in pixel-level dense prediction tasks in future.
>
> ## W3: Given that Lite DETR achieves comparable performance, the advantages of Dynamic DETR in real-world applications should be further clarified
>
> **Response**:
> First of all, we are sorry for the incorrect parameter descrition about DINO and Dynamic DINO in the initial submission, while the corrected verison is as follows. Our anonymous code please refers to [here](https://anonymous.4open.science/r/Dynamic-DETR-4D7F)
>
> Tab. T4. Corrected efficiency of DINO, Lite DINO, and Dynamic DINO on the COCO val-set.
> |Model|AP|AP$_{\mathrm{50}}$|AP$_{\mathrm{75}}$|FLOPs (G)|FPS|
> |-|:-:|:-:|:-:|:-:|:-:|
> |DINO|50.9|68.9|55.3|244.5|14.4|14.4|
> |Lite DINO|**50.4**|-|54.6|151.0|**23.2**|
> |Dynamic DINO|50.2|69.2|**54.7**|**141.7**|**23.2**|
> ||
>
> For COCO, our method slightly lead Lite DETR. However, on the LVIS and VOC datasets and when using MobileNet as the backbone, our method significantly outperforms Lite DETR in terms of AP, Params, and FPS. Noting that Lite DETR suffers from performance degradation under shorter training schedules, showing the efficiency of our dynamic strategy in reducing training costs.
>
> Moreover, to verify the potential of our dynamic token sparsification in real-time detection tasks, we investigate the performance of several efficient detectors when integrated a lightweight backbone MobileNet-V2, and the results are shown in **Tab. T5**.
>
> Tab. T5. Performance of DINO and various efficient solutions with  MobileNet-V2 on the COCO val-set, where the output channels are set to 256 for convergence.
> |Model|AP|AP$_{\mathrm{50}}$|AP$_{\mathrm{75}}$|FLOPs (G)|FPS|
> |-|:-:|:-:|:-:|:-:|:-:|
> |DINO|25.9|38.7|27.5|172.6|24.0|
> |Sparse DINO|19.8|33.8|20.7|67.5|27.2|
> |Lite DINO|18.8|29.2|19.9|78.4|**36.2**|
> |Focus DINO|20.0|31.8|20.7|79.8|24.7|
> |Dynamic DINO|**21.7**|**33.1**|**23.1**|**61.5**|36.0|
> ||
>
> Dynamic DINO significantly boosts the inference performance from 24.0 to 36.0 while the accuracy also outperforms its counterparts by a large margin. This competitive result shows the potential adaptability of our method for practical scenarios.

---

> > ### Comment · Reviewer_2gMK · 2025-04-06
> >
> > I appreciate the thorough reply from the authors. The majority of my questions have been clarified. The results on the LVIS and VOC datasets, in particular, provide strong evidence of the advantages of Dynamic DINO compared to Lite DINO. I am therefore inclined to slightly increase my score.

---

> > > ### Author Response · Authors · 2025-04-07
> > >
> > > We extend our sincere gratitude to the reviewer for the valuable time and insightful feedback. In the revised version, we will incorporate the breakdown of the latency analysis, results on additional datasets, and further discussions on the advantages of Dynamic DETR in real-world applications. Once again, we truly appreciate your thoughtful comments and recognition.

---

### Official Review · Reviewer_6br4 · 2025-03-12

**Overall Recommendation:** 3

**Summary:**

It is known in the object detection literature that the detection transformers are notorious for their long and computationally demanding training requirements. To partially address this issue, this work proposes a novel token aggregation strategy for detection transformers based on the recent token merging strategies. In particular, the work aims to exploit the redundancy of the tokens at different feature levels dynamically by merging them. Experimental results on COCO2017 aim to highlight the efficiency gains of the proposed method.

**Claims And Evidence:**

The main motivation behind the proposed method is motivated well and discussed in a detailed and objective manner. It also follows the established token merging literature [A, B, C] for computer vision.

Main claims of the work could be listed as follows:

**1.** Dynamic DETR performs at least on-par with the base models and other efficient DETR varieties.

**2.** Dynamic DETR consistently outperforms the efficiency of the models compared to the base models and other efficient DETR varieties.

**3.** Dynamic DETR framework finds _the_ sweet spot between the performance and efficiency.


With respect to these claims, the authors present:

**1.** The authors provide experimental results on COCO2017 _minival-set_. In these results, it is evident that the Dynamic DETR is almost always ~1 AP behind the base model, while performing on-par with Lite DETR [D] and Focus-attention DETR [E] in both of the presented settings.

**2.** The authors provide the FLOPs on COCO2017  _minival-set_. In these results, it is clear that the Dynamic DETR is more efficient with respect to FLOPs compared to base models, while being almost the same as Lite DETR [D] on D-DETR baseline and improving Lite DETR [D]  by 3% on the DINO baseline (albeit the FPS gain is almost the same as Lite DETR [D]).

**3.** Based on the aforementioned two results, Dynamic DETR slightly compromises the performance while providing reasonable efficiency improvements compared to the base models.

Following from these claims and presented evidence, it can be seen that Dynamic DETR is not an objectively better method compared to Lite DETR [D] (D-DETR efficiency results and DINO performance results on Table 1). In addition, while I acknowledge that the earlier works (e.g [A]) were also constrained to the FLOPs, the discussions in the work are limited solely with FLOPs, while the broader discussion on efficiency, and finding the sweet spot between efficiency and performance is a multi-faceted discussion, involving both number of training iterations and memory costs.

Based on these discussions, the claims of the work are not strongly supported.



[A] Bolya, Daniel, et al. "Token Merging: Your ViT But Faster." The Eleventh International Conference on Learning Representations.

[B] Bolya, Daniel, and Judy Hoffman. "Token merging for fast stable diffusion." Proceedings of the IEEE/CVF conference on computer vision and pattern recognition. 2023.

[C] Yuan, Xin, Hongliang Fei, and Jinoo Baek. "Efficient transformer adaptation with soft token merging." Proceedings of the IEEE/CVF Conference on Computer Vision and Pattern Recognition. 2024.

[D] Li, Feng, et al. "Lite detr: An interleaved multi-scale encoder for efficient detr." Proceedings of the IEEE/CVF conference on computer vision and pattern recognition. 2023.

[E] Zheng, Dehua, et al. "Less is more: Focus attention for efficient detr." Proceedings of the IEEE/CVF international conference on computer vision. 2023.

**Essential References Not Discussed:**

N/A

**Experimental Designs Or Analyses:**

Other than the aforementioned issue on the usage of minival-test, the rationale behind it and a short description of exactly what it corresponds to, the experiments seem sound.

**Methods And Evaluation Criteria:**

- Basing the experiments on COCO2017 is valid as it is frankly the most established object detection benchmark. However, the rationale behind the usage of minival-set is not very clear.

- In addition, object detectors (and thus DETRs) are utilized in broad range of domains, often involving long-tailed and challenging cases. Therefore, performing analyses/discussions on other established datasets involving much more classes and much denser annotations, such as LVIS [A] could be helpful for the work. From a token merging point-of-view, these cases could be more challenging given the dense nature of object annotations, though it would also be impressive Dynamic DETR works on it too.

**Other Comments Or Suggestions:**

- There is a minor typo on Pg.5: hyperparamt -> hyper parameter
- Some figures, such as Figure 1 have very small font size. They are otherwise carefully designed and neat.

**Other Strengths And Weaknesses:**

The idea proposed in the work is an extension of token merging for the detection literature. It contains various novel design choices to make this extension work in an efficient and reasonably performant manner. The writing is also mostly clear, although the flow of thought was a bit counter-intuitive for me between Sections 3.2 and 3.3, since the exact definitions of concepts in 3.2 are defined later.

**Questions For Authors:**

- Where do you think your work stands with respect to other efficiency concerns, such as the number of training iterations and memory imprint?
- Do you think your work is complementary, synergic or orthogonal to other efficient DETR methods, such as [A] from ECCV 2024?

[A] Yavuz, Feyza, et al. "Bucketed Ranking-Based Losses for Efficient Training of Object Detectors." European Conference on Computer Vision. Cham: Springer Nature Switzerland, 2024.

**Relation To Broader Scientific Literature:**

Object detection is one of the primary and most well-established computer vision tasks. Detection transformers are the trailblazing response of the detection community to the surge of transformers, though they are known to have various efficiency issues. Thus, the scope of the work is interesting to the broader audiences of detection and efficiency communities.

**Theoretical Claims:**

The nature of the paper is mostly empirical without detailed theoretical claims. However, the design choices made in the paper are mostly motivated and explained by relevant examples and visualizations.

---

> ### Author Rebuttal · Authors · 2025-04-01
>
> Thanks for your thoughtful comments. Below we response to these concerns.
>
> ## C1: Comparable performance with Lite DETR
>
> **Response**:
> First of all, we are sorry for the incorrect parameter descrition about DINO and Dynamic DINO in the initial submission, while the corrected verison is as follows. Our anonymous code please refers to [here](https://anonymous.4open.science/r/Dynamic-DETR-4D7F)
>
> Tab. T1. Corrected efficiency of DINO, Lite DINO, and Dynamic DINO on the COCO val-set.
> |Model|AP|FLOPs (G)|FPS|
> |-|:-:|:-:|:-:|
> |DINO|50.9|244.5|14.4|
> |Lite DINO|**50.4**|151.0|**23.2**|
> |Dynamic DINO|50.2|**141.7**|**23.2**|
> ||
>
> To throughly explore the advantages of our Dynamic strategy to Lite DETR, we take DINO as the baseline detector and compare the performance between these two methods as follows.
> Our method outperforms Lite-DETR across different datasets and backbone networks, with even greater advantages under a shorter training schedule, further demonstrating its efficiency.
>
>  - On COCO val-set, Swin-T, 12 epochs.
> |Model|AP|FLOPs (G)|FPS|GPU Memory (G)|Training Hours (h:m)|
> |-|:-:|:-:|:-:|:-:|:-:|
> |DINO|51.5|252.3|14.0|27892|**23:27**|
> |Lite DINO|48.3|151.0|16.8|**26371**|24:20|
> |Dynamic DINO|**49.9**|**149.4**|**18.2**|27764|23:48|
> ||
>
>  - On LVIS val-set, ResNet-50, 12 epochs.
> |Model|AP|FLOPs (G)|FPS|GPU Memory (G)|Training Hours (h:m)|
> |-|:-:|:-:|:-:|:-:|:-:|
> |DINO|26.1|247.1|19.8|24244|14:59|
> |Lite DINO|20.2|160.0|16.0|42348|13:48|
> |Dynamic DINO|**23.4**|**146.6**|**22.5**|**22557**|**12:15**|
> ||
>
>  - On COCO val-set, MobileNet-V2, 12 epochs.
> |Model|AP|FLOPs (G)|FPS|GPU Memory (G)|
> |-|:-:|:-:|:-:|:-:|
> |DINO|25.9|172.6|24.0|15304|
> |Lite DINO|18.8|78.4|**36.2**|33950|
> |Dynamic DINO|**21.7**|**61.5**|36.0|**14082**|
> ||
>
>  - On VOC2007 val-set, ResNet-50, 12 epochs.
> |Model|AP|FLOPs (G)|FPS|
> |-|:-:|:-:|:-:|
> |DINO|65.7|241.6|15.5|
> |Lite DINO|38.1|151.0|**21.3**|
> |Dynamic DINO|**63.8**|**135.2**|21.1|
> ||
>
> ## C2: The usage of minival-set
>
> **Response**: Sorry for our unclear descriptions. All the results in our paper are evaluated on COCO val-set, which is strictly consistent with previous works (*Sparse detr, Lite detr, and Focus detr*).
>
> ## C3: Experiemnts on LVIS dataset
>
> **Response**: We perform experiments on LVIS datasets to verify the generalizability and robustness of our approach.
>
> Tab. T2. Performance of DINO and various efficient solutions on the LVIS-1.0 val-set.
> | Model | AP  | AP$_{\mathrm{50}}$  | AP$_{\mathrm{75}}$  | AP$_{\mathrm{r}}$  | AP$_{\mathrm{c}}$  | AP$_{\mathrm{f}}$   | FLOPs (G) | FPS  | GPU Memory (G) |
> |-|:-:|:-:|:-:|:-:|:-:|:-:|:-:|:-:|:-:|
> | DINO| 26.1 | 34.5 | 27.5 | 8.3 | 24.1 | 36.1 | 247.1 | 19.8 | 24244|
> | Sparse DINO| 22.9 | 32.0 | 24.2 | 8.4 | 21.3 | 30.9 | 151.7 | 21.2 | 22680|
> | Lite DINO| 20.2 | 28.0 | 21.4 | 3.0 | 17.5 | 30.8 | 160.0 | 16.0 | 42348|
> | Focus DINO| **23.7** | **32.9** | **25.2** | **10.2** | **21.7** | 31.9 | 168.2 | 20.4 | 22640|
> | Dynamic DINO| 23.4 | 31.8 | 25.0 | 7.7 | 20.8 | **33.4** | **146.6** | **22.5**| **22557**|
> ||
>
> As exhibited in **Tab. T3**, our Dynamic DINO lags Focus DINO slightly by 0.3% points, but outperforms it in inference speed by 2.1 FPS. To sum up, the proposed Dynamic token aggregation strategy significantly reduces the parameters of the baseline model (DINO), while also exhibiting a smaller performance loss compared to other efficient solutions.
>
> ## C4: Writing, typos and figures
>
> **Response**: We sincerely appreciate your detailed comments on writing and figure clarity.
> We will clarify the flow between Sections 3.2 and 3.3 to ensure a more intuitive progression of ideas. Moreover, the typos will be corrected and the font size in Figure 1 will be enlarged for better readability.
>
> ## C5: Comparison between ours and other competitors in training iterations and memory imprint
>
> **Response**: Considering the training iterations of these methods are similar, we report the memory info of several setups (see **Tab. T2, T3, and T4** and **the responses to C1**). In comparison to other counterparts, our method shows best overall performance.
>
> Tab. T3. Swin-Transformer on the COCO val-set.
> |Model|AP|FLOPs (G) | FPS  | GPU Memory (G) |
> |-|:-:|:-:|:-:|:-:|
> |DINO|51.5|252.3|14.0|27892|
> |Sparse DINO|49.6|137.0|18.0|**24696**|
> |Lite DINO|48.3|151.0|16.8|26371|
> |Focus DINO|**49.9**|156.9|15.3|31933|
> |Dynamic DINO|**49.9**|**149.4**|**18.2**|27764|
> ||
>
> Tab. T4. MobileNet-V2 on the COCO val-set.
> |Model|AP|FLOPs (G)|FPS|GPU Memory (G) |
> |-|:-:|:-:|:-:|:-:|
> |DINO|25.9|172.6|24.0|15304|
> |Sparse DINO|19.8|67.5|27.2|15810|
> |Lite DINO|18.8|78.4|**36.2**|33950|
> |Focus DINO|20.0|79.8|24.7|14141|
> |Dynamic DINO|**21.7**|**61.5**|36.0|**14082**|
> ||
>
> ## C6: Integration with BR Loss
>
> **Response**: The BR Loss could consitently facilitate the final performance of both DINO and Dy-DINO, see **Tab. T5**.
>
> Tab. T5. Swin-Transformer on the COCO val-set.
> | Model|DINO|DINO+BR|Dy-DINO|Dy-DINO+BR|
> |-|:-:|:-:|:-:|:-:|
> |AP|51.5|52.2|49.9|50.3|
> ||

---

> > ### Comment · Reviewer_6br4 · 2025-04-05
> >
> > I appreciate the detailed response of the authors to the issues I raised. In particular, the authors have provided further empirical evidence related to how Dynamic DETR compares against Lite DETR where they have shown that it indeed provides non-trivial gains over more settings. In addition, the authors have provided detailed discussions related to different efficiency measures, including not FLOPs this time but also memory imprints.
> >
> > Based on these points, I will be raising my score accordingly.

---

> > > ### Author Response · Authors · 2025-04-06
> > >
> > > Thank you very much for your thoughtful reconsideration and for your encouraging comments. We're glad that our additional analysis and clarifications addressed your concerns, and we will incorporate these additions into the revised version. Again, we truly appreciate your willingness to engage deeply with our work.

---

### Official Review · Reviewer_sTgo · 2025-03-13

**Overall Recommendation:** 3

**Summary:**

The paper **"Not All Tokens Matter All The Time: Dynamic Token Aggregation Towards Efficient Detection Transformers"** proposes a novel framework called **Dynamic DETR**, aiming to address the computational efficiency bottleneck in **Detection Transformers (DETRs)**. DETRs require high computational resources, especially in the encoder, which becomes a major bottleneck. Existing methods generally adopt **static token sparsification strategies**, ignoring the differences in token importance across different layers and encoder blocks, leading to performance degradation. Dynamic DETR significantly reduces computational costs while maintaining high detection accuracy by dynamically adjusting token density and incorporating a multi-level token sparsification strategy. The main contributions include:

**Dynamic Token Aggregation:** Dynamically adjusts token density to reduce redundancy and computational complexity.

**Multi-level Token Aggregation:** Employs neighbor-based aggregation at lower levels to preserve spatial details and global aggregation at higher levels to capture contextual information.

**Representational Center-distance Regularization:** Ensures consistency in feature distribution before and after sparsification through regularization, improving detection performance.

**Claims And Evidence:**

The claims made in the paper are well-supported by extensive experiments. The authors conducted experiments on the **COCO2017 dataset**, demonstrating the effectiveness of **Dynamic DETR**. The results show that **Dynamic DETR significantly reduces FLOPs (by 39.7%-53.9%)** across multiple DETR variants while incurring only **a slight performance drop (AP decrease of 0.5%-1.0%)**. Additionally, ablation studies confirm the effectiveness of dynamic token aggregation, multi-level aggregation strategies, and representational center-distance regularization.

**Essential References Not Discussed:**

The paper cites a large number of relevant references covering DETR variants and token sparsification algorithms. No critical missing references were identified.

**Experimental Designs Or Analyses:**

The experimental design and analyses are reasonable and effective. The authors evaluate Dynamic DETR on **multiple DETR variants**, demonstrating its **generalizability and effectiveness**. Additionally, ablation studies confirm the contribution of each component, ensuring the reliability of the experimental results.

**Methods And Evaluation Criteria:**

The proposed methods and evaluation criteria are reasonable and well-suited for object detection tasks. **Dynamic DETR** effectively reduces redundant tokens while maintaining detection accuracy through dynamic token density adjustment and multi-level aggregation. The experiments are conducted on the **COCO2017 dataset**, with evaluation metrics including **AP, AP50, and AP75**, ensuring comprehensive assessment.

**Other Comments Or Suggestions:**

- The experimental results and analyses are comprehensive, but the authors could further discuss **the potential of Dynamic DETR in practical applications.**

- It is suggested that the authors explore **Dynamic DETR's applicability in other Transformer architectures** in future work to verify its generalizability.

**Other Strengths And Weaknesses:**

**strengths:**

- **Dynamic DETR effectively reduces computational costs** while maintaining high detection accuracy by dynamically adjusting token density.

- **Multi-level token aggregation strategy and representational center-distance regularization** enhance model robustness and generalization ability.

- **Comprehensive experimental design**, validating the method's effectiveness and generalizability across multiple DETR variants.

**Weaknesses:**

- The paper **does not discuss the real-world deployment performance** of Dynamic DETR, such as its effectiveness in **real-time detection tasks.**

- Although **Dynamic DETR performs well on COCO**, its performance on **other datasets (e.g., Pascal VOC) remains unverified.**

**Questions For Authors:**

1. **How does Dynamic DETR perform in real-world applications, such as real-time detection tasks? Are there any plans to conduct related experiments?**

    - **Impact**: Understanding Dynamic DETR’s real-world performance helps assess its deployment potential.

2. **How does Dynamic DETR perform on other datasets (e.g., Pascal VOC)? Are there plans for cross-dataset validation?**

    - **Impact**: Cross-dataset validation can further demonstrate the generalizability and robustness of Dynamic DETR.

3. **Is Dynamic DETR applicable to other Transformer architectures (e.g., ViT)? Are there plans to explore this aspect?**

    - **Impact**: Investigating Dynamic DETR’s applicability in other Transformer architectures can further verify its **generalizability and scalability.**

**Relation To Broader Scientific Literature:**

This work is closely related to existing **DETR improvement methods** and **token sparsification algorithms.** Existing DETR variants (e.g., **Deformable DETR, DAB-DETR**) improve detection performance but still have **high computational costs**. **Dynamic DETR** addresses this limitation by **introducing a dynamic token aggregation strategy**, significantly reducing computational costs while maintaining high detection accuracy, filling the gap left by previous methods.

**Theoretical Claims:**

The paper does not propose theoretical proofs, so there is no need to verify the correctness of theoretical claims.

---

> ### Author Rebuttal · Authors · 2025-03-31
>
> We sincerely appreciate your selfless dedication and thoughtful comments. Below we response to these concerns.
>
> ## W1&Q1: The paper does not discuss the real-world deployment performance of Dynamic DETR, such as its effectiveness in real-time detection tasks
>
> **Response**:
> To verify the potential of our dynamic token aggregation in real-time detection, we investigate the performance of several efficient detectors when integrated a lightweight backbone MobileNet-V2, and the results are shown in **Tab. T1**.
>
> Tab. T1. Performance of DINO and various efficient solutions with MobileNet-V2 on the COCO val-set, where the output channels are set to 256 for convergence and trained for 12 epochs.
> |Model|AP|AP$_{\mathrm{50}}$|AP$_{\mathrm{75}}$|FLOPs (G)|FPS|
> |-|:-:|:-:|:-:|:-:|:-:|
> |DINO|25.9|38.7|27.5|172.6|24.0|
> |Sparse DINO|19.8|33.8|20.7|67.5|27.2|
> |Lite DINO|18.8|29.2|19.9|78.4|**36.2**|
> |Focus DINO|20.0|31.8|20.7|79.8|24.7|
> |Dynamic DINO|**21.7**|**33.1**|**23.1**|**61.5**|36.0|
> ||
>
> Dynamic DINO significantly boosts the inference performance from 24.0 to 36.0 while the accuracy also outperforms its counterparts by a large margin. This competitive result shows the potential adaptability of our method for practical scenarios.
>
> Moreover, to enable a fair and comprehensive comparison, our experiment setups strictly align with pioneering works [R1, R2, R3], which primarily focus on benchmark evaluations and do not explicitly consider deployment performance. Future work could explore lightweight adaptations and hardware-friendly implementations to further bridge the gap between research and practical deployment.
> - [R1] Roh, B., et al. Sparse detr: Efficient end-to-end object detection with learnable sparsity. ICLR, 2022.
> - [R2] Li, Feng, et al. Lite detr: An interleaved multi-scale encoder for efficient detr. CVPR, 2023.
> - [R3] Zheng, Dehua, et al. Less is more: Focus attention for efficient detr. ICCV, 2023.
>
> ## W2&Q2: Although Dynamic DETR performs well on COCO, its performance on other datasets (e.g., Pascal VOC) remains unverified.
>
> **Response**: To demonstrate the generalizability and robustness of our Dynamic DETR, as suggested by your valuable comment, we perform experiments on VOC and LVIS datasets, two of the most commonly used benchmarks beyond COCO for object detection. The results are as exhibited as **Tab. T2** and **Tab. T3**. Note that all the models are with a ResNet-50 as the backbone and trained for a bunch of 12 epochs.
>
> Tab. T2. Performance of DINO and various efficient solutions on the VOC2007 val-set.
> | Model| mAP|FLOPs (G)|FPS|
> |-|:-:|:-:|:-:|
> | DINO|65.7| 241.6|15.5|
> | Sparse DINO|62.5|141.4|19.6|
> | Lite DINO |38.1|151.0|**21.3**|
> | Focus DINO|51.4| 153.6|20.2|
> | Dynamic DINO|**63.8**|**135.2**|21.1|
> ||
>
> Tab. T3. Performance of DINO and various efficient solutions on the LVIS-1.0 val-set.
> | Model | AP  | AP$_{\mathrm{50}}$  | AP$_{\mathrm{75}}$  | AP$_{\mathrm{r}}$  | AP$_{\mathrm{c}}$  | AP$_{\mathrm{f}}$   | FLOPs (G) | FPS  |
> |-|:-:|:-:|:-:|:-:|:-:|:-:|:-:|:-:|
> | DINO|26.1|34.5|27.5|8.3|24.1|36.1|247.1|19.8|
> | Sparse DINO| 22.9 | 32.0|24.2|8.4|21.3|30.9|151.7|21.2|
> | Lite DINO| 20.2 | 28.0|21.4|3.0|17.5|30.8|160.0|16.0|
> | Focus DINO| **23.7** | **32.9** | **25.2**|**10.2**|**21.7**|31.9|168.2|20.4|
> | Dynamic DINO| 23.4 | 31.8|25.0 |7.7|20.8|**33.4**|**146.6**|**22.5**|
> ||
>
> Consistent with the performance on the COCO dataset, the proposed dynamic token aggregation significantly reduces the parameters of the baseline model (DINO), while also exhibiting a smaller performance loss compared to other efficient solutions. Specifically, as exhibited in **Tab. T2**, our Dynamic DINO scores 63.8% AP on the VOC dataset, which is 1.9% points lower than the baseline DINO but with a 36% improvement in FPS, and excels other competitors by a large margin both in accuracy and efficiency. For the LVIS results in **Tab. T3**, our Dynamic DINO lags Focus DINO slightly by 0.3% points, but outperforms it in inference speed by 2.1 FPS.
>
> In summary, the results on VOC and LVIS datasets further showcase the superiority and generality of our Dynamic token sparisification strategy.
>
> ## Q3: Is Dynamic DETR applicable to other Transformer architectures (e.g., ViT)? Are there plans to explore this aspect?
>
> **Response**: To further explore the generalizability of Dynamic DETR, we conducted additional experiments using Swin Transformer as the backbone. The results in **Tab. T4** demonstrate that our approach remains effective across different Transformer architectures, highlighting its adaptability.
>
> Tab. T4. Performance of DINO and various efficient solutions with  Swin-Transformer on the COCO val-set.
> | Model | AP  | AP$_{\mathrm{50}}$  | AP$_{\mathrm{75}}$|FLOPs (G) | FPS  |
> |-|:-:|:-:|:-:|:-:|:-:|
> |DINO|51.5|70.2|56.5|252.3|14.0|
> |Sparse DINO|49.6|68.4|54.1|137.0|18.0|
> |Lite DINO|48.3|66.1|52.8|151.0|16.8|
> |Focus DINO|**49.9**|68.2|54.3|156.9|15.3|
> |Dynamic DINO|**49.9**|68.8|54.3|**149.4**|**18.2**|
> ||

---

> > ### Comment · Reviewer_sTgo · 2025-04-06
> >
> > Thank you for the detailed response. You have conducted substantial justification and experiments, and I have raised my score accordingly.

---

> > > ### Author Response · Authors · 2025-04-07
> > >
> > > Thank you very much for your recognition and for taking the time to go through our work in detail. We're glad that the additional justifications and experiments helped clarify our approach. We sincerely appreciate your revised score and constructive feedback!

---

### Decision · Program_Chairs · 2025-05-01

**Decision:**

Accept (poster)

**Comment:**

The paper addresses the important problem of efficiency in Detection Transformers. While the initial reviews raised valid concerns regarding the scope of evaluation and the demonstrated advantage over existing methods, the authors provided an exceptionally thorough rebuttal with significant new experimental results. These results convincingly demonstrate the method's effectiveness and generalizability across multiple datasets (COCO, VOC, LVIS) and backbones (ResNet, Swin-T, MobileNet-V2), often showing favourable trade-offs compared to strong baselines like Lite DETR. The additional analyses on memory, training time, and latency further strengthen the paper. The reviewers were largely convinced by the rebuttal, leading to score increases. The proposed dynamic, multi-level token aggregation strategy combined with representation regularization offers a solid contribution to the field of efficient object detection. Assuming the authors incorporate the promised revisions regarding clarity and integrate the new results into the final paper, it is recommended for acceptance.